# A genomic lifespan program that reorganises the young adult brain is targeted in schizophrenia

Nathan G Skene, Marcia Roy, Seth GN Grant*

Genes to Cognition Programme, Centre for Clinical Brain Sciences, University of Edinburgh, Edinburgh, United Kingdom

**Abstract** The genetic mechanisms regulating the brain and behaviour across the lifespan are poorly understood. We found that lifespan transcriptome trajectories describe a calendar of gene regulatory events in the brain of humans and mice. Transcriptome trajectories defined a sequence of gene expression changes in neuronal, glial and endothelial cell-types, which enabled prediction of age from tissue samples. A major lifespan landmark was the peak change in trajectories occurring in humans at 26 years and in mice at 5 months of age. This species-conserved peak was delayed in females and marked a reorganization of expression of synaptic and schizophrenia-susceptibility genes. The lifespan calendar predicted the characteristic age of onset in young adults and sex differences in schizophrenia. We propose a genomic program generates a lifespan calendar of gene regulation that times age-dependent molecular organization of the brain and mutations that interrupt the program in young adults cause schizophrenia.

DOI: https://doi.org/10.7554/eLife.17915.001

## Introduction

Identifying the genetic mechanisms that underpin brain ageing across the lifespan may provide explanations for the maturation of behaviours and age of onset of diseases. Longitudinal studies show cognition, emotion and personality emerge progressively during childhood and adolescence, with executive functions peaking in early adulthood (*Craik and Bialystok, 2006*; *De Luca et al., 2003*). This coincides with the onset of some of the most devastating psychiatric disorders, most of which arise during later stages of brain development in the teenage years and early twenties (*Kessler et al., 2007*). For example, impulse-control disorders arise in late childhood and early teen years, substance abuse peaks in the early twenties, and schizophrenia in the mid-twenties (with a delay of around two years in females) (*Häfner et al., 1993*). Some monogenic neurological disorders also have early adult onset, such as Inclusion Body Myopathy associated with Paget disease Fronto-temporal Dementia (*Watts et al., 2004*) and rapid-onset dystonia Parkinsonism (*Brashear et al., 2012*).

Why some brain disorders with a strong genetic component have a late developmental onset is unknown. The prevailing hypothesis for schizophrenia proposes that an early (fetal) insult or mutation renders the brain vulnerable to a secondary environmental insult that occurs in young adults, which then triggers the onset of psychosis (*Bayer et al., 1999*). However, the finding that the age of onset for schizophrenia has a heritability estimated at 33% (*Hare et al., 2010*), suggests that the timing may have a genetic basis. In recent years, there has been major progress in understanding the genetic basis of schizophrenia with the identification of many mutations and variants contributing to disease susceptibility. It is widely accepted that many mutations directly impact on synapse proteins, particularly those involved with postsynaptic signalling mechanisms in excitatory synapses (*Kirov et al., 2008*; *Pocklington et al., 2015*; *Fromer et al., 2014*; *Fernández et al., 2009a*;

*For correspondence: seth.
grant@ed.ac.uk

Competing interests: The authors declare that no competing interests exist.

**eLife digest** In our lifetime, we go through many changes – physically and also intellectually. At certain ages, we are particularly vulnerable to develop psychiatric disorders, and the majority of mental conditions start to manifest in teenagers and young adults. The symptoms for schizophrenia, for example, a mental health disorder in which patients often experience hallucinations, delusion or changes in behavior, typically start in the mid-twenties.

Schizophrenia tends to run in families and it is likely that different combinations of faulty genes that affect the connections between nerve cells increase the chance of having the disease. Until now, scientists have assumed that certain situations and environmental factors trigger the condition, but it was unknown if genes could influence the age at which the disease will begin.

To explore whether genes in the brain change at certain time points, Skene et al. examined how the genes are turned on and off across the lifespan of healthy mice and humans. The results showed that in both mice and humans, a 'genetic lifespan calendar' controlled every cell type in the brain and directed the way they worked at different ages. The timing was so precise that it was possible tell the age of a mouse or a person simply by looking at the way the genes were expressed in a tissue sample.

Skene et al. then studied how the genetic lifespan calendar controlled the genes damaged in schizophrenia, and found that the calendar caused a major reorganization of the genes at the time when the symptoms started. This suggests that the genetic lifespan calendar is a crucial factor that can determine at what age the disease will start.

The next step will be to study how the genetic lifespan calendar programs changes throughout the brain and to explore if it could be manipulated to change how the brain ages. This could help to develop new types of treatments for schizophrenia and other conditions of the brain.
DOI: https://doi.org/10.7554/eLife.17915.002

*Singh et al., 2017*). The postsynaptic proteome of excitatory synapses is physically organised into multiprotein complexes of which the supercomplexes assembled by PSD95 (*Husi et al., 2000*; *Frank et al., 2016a*; *Frank and Grant, 2017*; *Frank et al., 2017*) play a major role in regulating cognitive functions (*Migaud et al., 1998*; *Nithianantharajah et al., 2013*) and are disrupted by schizophrenia mutations (*Pocklington et al., 2015*; *Fromer et al., 2014*; *Fernández et al., 2009a*; *Singh et al., 2017*; *Kirov et al., 2012*; *Purcell et al., 2014*). Together these observations suggest there may be genetic mechanisms that account for the convergence between the many schizophrenia susceptibility genes, the postsynaptic proteome and the young adult brain.

Transcriptome profiling of the brain at different ages has shown complex changes in expression levels across the lifespan (*Colantuoni et al., 2011*). In early onset brain disorders, such as intellectual disability and autism, gene expression levels in fetal and early postnatal development correlate with enrichment in autism susceptibility genes (*Willsey et al., 2013*; *Parikshak et al., 2013*). However, correlation based approaches (*Willsey et al., 2013*; *Parikshak et al., 2013*; *Gulsuner et al., 2013*) (e.g. weighted gene coexpression network analysis) are unable to account for the full complexity of transcriptional changes that occur with age. Furthermore, although there has been extensive characterisation of cellular and anatomical maturation in neuronal, glial and synaptic subtypes (*Alexander and Goldman, 1978*; *Shaw et al., 2006*; *Gogtay et al., 2004*, *2006*; *Zehr et al., 2006*; *Woo et al., 1997*; *Huttenlocher, 1979*; *Anderson et al., 1995*; *Bourgeois et al., 1994*; *Kalsbeek et al., 1988*; *Klingberg et al., 1999*), the relevant transcriptome changes remain to be identified.

To further understand the transcriptional events underlying brain development and ageing, we developed new tools that identify age-dependent gene regulatory events. These methods detect when gene expression trajectories change direction or plateau: we refer to these events as Transcriptome Trajectory Turning Points (TTTPs) (*Figure 1*). The timing of TTTPs has been shown to be an important feature of transcriptome trajectories during maize embryo development and the yeast cell-cycle: in both systems, genes with linked biological functions were found to turn/plateau at similar time points (*Lee et al., 2002*; *Spellman et al., 1998*). We have characterised TTTPs in the neocortex of humans and hippocampus of mice across their respective postnatal lifespans. This revealed a

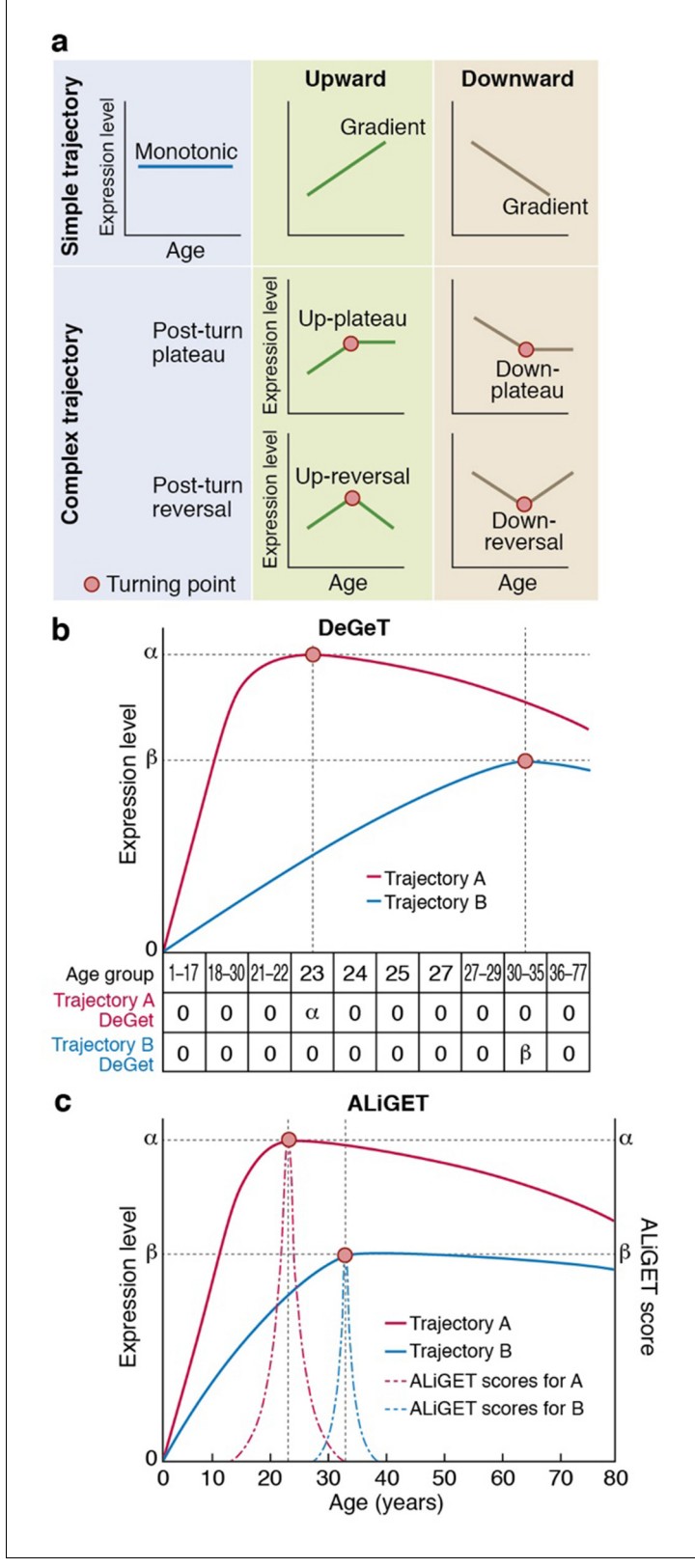

**Figure 1.** Gene expression trajectories can be classified and quantified based on the characteristics of their turning points (TTTPs). (a) Simple trajectories (upper panel) do not contain TTTPs (red dots) whereas complex trajectories (lower panel) contain a TTTP. Complex trajectories are further classified according into those that plateau or reverse direction after the TTTP. Upward and Downward classification refers to the initial direction of

*Figure 1 continued on next page*

*Figure 1 continued*

the trajectory. (**b**) and (**c**). Examples of the application of the DeGeT (**b**) and ALiGeT (**c**) methods applied to two trajectories (A, shown in red; turns at 23 years of age and has a greater fold change than B, shown in blue, which turns at 33 years). The change in expression (ΔE) prior to the turning point is denoted as α and β for trajectories A and B respectively. (**b**) The DeGeT scoring system generates a score for each of 10 groups of ages, where each group contains a similar number of TTTPs. Trajectory A receives a DeGeT score of α in the age group spanning 23 years and zero in all other ages. Similarly, trajectory B receives a score of β only in age group 30–35. (**c**) The ALiGeT scoring system generate a score for each year of age and decays the contribution of ΔE as a function of distance from the TTTP (see dotted ALiGeT scores that peak at α and β and rapidly decay at ages either side of this peak).

DOI: https://doi.org/10.7554/eLife.17915.003

The following figure supplement is available for figure 1:

**Figure supplement 1.** ALiGeT scoring assigns a score to each gene for each year of age.
DOI: https://doi.org/10.7554/eLife.17915.004

previously unknown, species conserved, gene regulatory program. These methods were also used with single-cell transcriptomes to define the age-dependent sequence of changes in neuronal, glial and endothelial cell-types. Our data suggest that the late onset of some psychiatric and neurological disorders is timed by mutations in this genetically programmed developmental sequence. Our findings also indicate that misregulation in the molecular maturation of synapse proteomes during a critical window in young adults is important for the onset of schizophrenia. These methods and findings open new areas of investigation into the genetic regulation of brain age and highlight their importance in the adolescent and young adult.

## Results

### Quantifying expression trajectories

As shown in the schematic diagram (*Figure 1a*), genes can be categorised according to their age-dependent profile of expression into simple trajectories (monotonic, upward or downward gradients) or complex trajectories that contain a Trancriptome Trajectory Turning Point (TTTP), marking both the direction of change and the age at which it occurs. Following a TTTP, the expression may reverse direction or plateau. We systematically identified gene expression trajectories by fitting cubic splines and marking TTTPs where the first derivative dE/dA, (E = expression level and A = Age) of the interpolated trajectories equals zero and changes sign. Next, to take into account the extent of expression changes prior to the TTTP (ΔE), and thereby emphasise those genes for which the TTTP represents a significant regulatory event, we developed two complementary methods (DeGeT, Decile-based Gene Turning; ALiGeT, Age-Linked Gene Turning). The DeGeT method depicted in *Figure 1b* is conceptually the simplest: the lifespan is divided into ten age groups within which an approximately equal number of TTTPs occur; each gene then receives a score for each age group (ΔE if the gene turns within that age window, and zero otherwise). The ALiGeT method depicted in *Figure 1c* extends this to generate a score for each year of age, by decaying the contribution of ΔE the greater the distance between the TTTP and the scored year (example trajectories and associated ALiGeT scores are shown in *Figure 1—figure supplement 1*). The reason for using two scoring methods is that some age periods have many TTTPs and others have few: DeGeT controls for this by balancing the number of TTTPs within age groups (and thereby has greater power to detect enrichments in earlier/later life stages), while ALiGeT scoring allows for the possibility that small time windows will have distinct molecular associations. Together, the TTTP, DeGeT and ALiGeT methods provide general purpose tools for exploring age-dependent gene regulation.

### Brain expression trajectories in human and mouse

We first applied these methods to the Braincloud dataset, which measured mRNA expression levels from 269 prefrontal cortex samples across the human lifespan (14th gestational week to 78 years). Although TTTPs were identified across the lifespan, they were sharply concentrated during early adulthood. Summing the number at each age shows a striking peak and a mean of 26.0 years in

males and 27.5 years in females (Wilcoxon signed rank test, p=0, *Figure 2a*, example trajectories with different turning points shown in *Figure 2—figure supplement 1*). Prominent peaks in early adulthood were confirmed with three regression methods (cubic splines with three degrees of freedom, four degrees of freedom and Loess regression) (*Figure 2b*). Removing X-chromosome genes from the analysis had no effect on this sex difference (p=0). To determine whether this TTTP-peak was human specific, we performed an equivalent analysis using transcriptome data from the hippocampus of 186 mice of both sexes between 58 and 600 days of age (*Figure 2—figure supplement 2*): this also revealed a peak in the frequency of TTTPs with a mean of 156 days for male and 165 days for female mice (p<$1\times10^{-323}$, *Figure 2c*). Although there is a lack of previous research on the age equivalence of early adulthood between rodents and humans, our results are concordant with estimations made using the TranslatingTime species comparison model which suggests that p156 (5 months) is equivalent to human early adulthood based on equivalent levels of cortical synaptic maturation (*Pinto et al., 2015*; *Pinto et al., 2013*; *Workman et al., 2013*). Plotting the cumulative distribution of TTTPs across the lifespan further illustrates the TTTP-peak in young adults (*Figure 2d,e*), and shows that 90% of TTTPs occur by 40 years of age, corresponding to the last stages of human brain development (*De Luca et al., 2003*; *Lebel et al., 2012*; *Wood et al., 2004*) and 7 months of age in mice. These data indicate that despite greatly differing lifespans, these two mammalian species share a lifespan program of brain gene expression with conserved features.

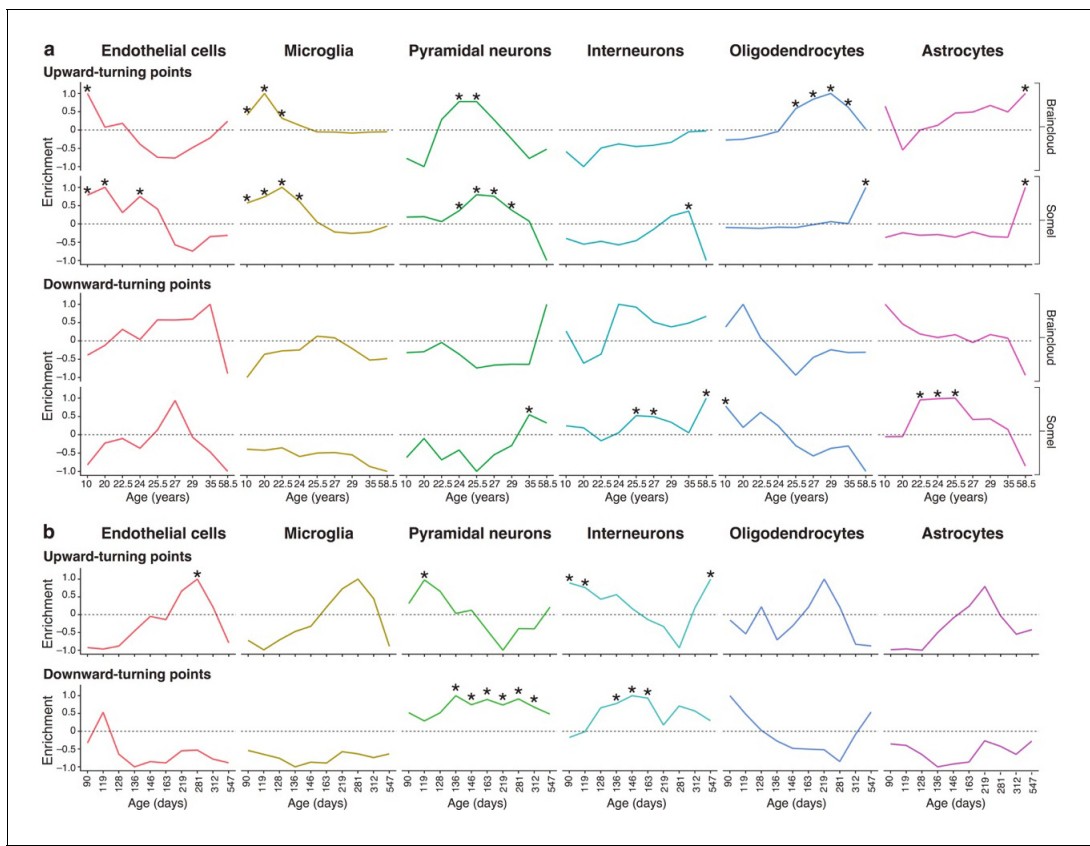

**Figure 3.** Genes associated with high levels of expression in particular cell types are enriched at different stages of the human and mouse lifespan. (**a**) Expression Weighted Cell-type Enrichments (EWCE) for the top 10% of genes with largest DeGeT scores within each age window for the human prefrontal cortex. Enrichments were calculated separately for the Somel (*Somel et al., 2009*) and Braincloud (*Colantuoni et al., 2011*) datasets. Enrichment values represent how far the mean cellular expression level within the target list is from the mean value for the bootstrapped lists (in terms of standard deviations); these values have been scaled to have a maximal value of either 1 or −1 within each age window. X-axis labels mark the mean age for each of the age windows. Asterisks mark those enrichments that are significant after Bonferroni correction with p<0.05. (**b**) Analysis as in (**a**) on mouse hippocampus.
DOI: https://doi.org/10.7554/eLife.17915.009

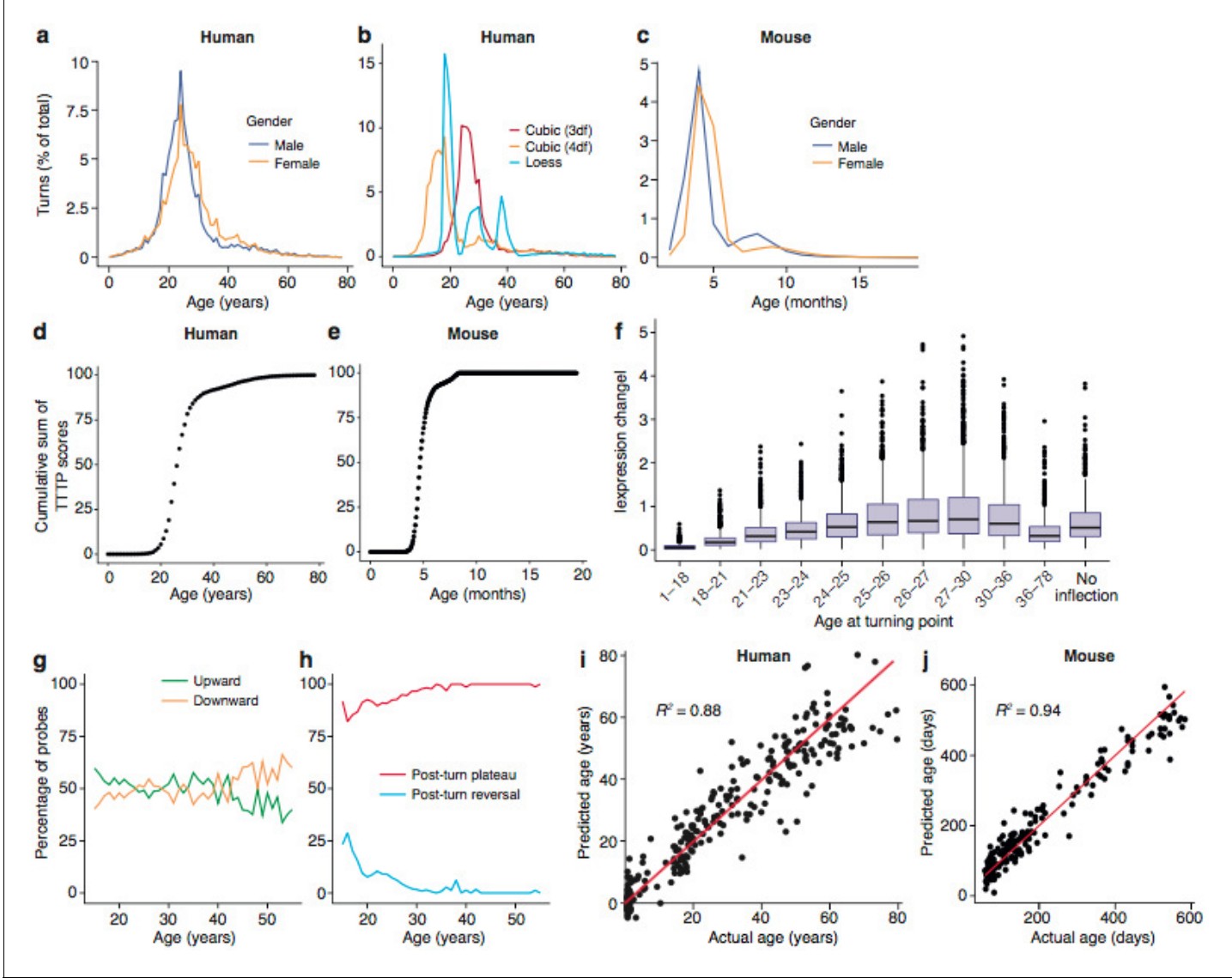

**Figure 2.** Trajectories and turning points characterise brain age. (**a**) Percentage of TTTPs at each age of human lifespan. Mean age is 26.0 years for males and 27.5 years for females. (**b**) Percentage of TTTPs at each age of human lifespan using three different methods for fitting splines. Mean age is 31.3 years using cubic splines with 3 degrees of freedom, 21.6 years using cubic splines with 4 degrees of freedom and 25.2 years using Loess regression. (**c**) Percentage of TTTPs at each age of mouse. Mean age is 156 days for males and 165 days for females. (**d**) Cumulative sum of TTTP scores for every year of life in the Braincloud dataset. (**e**) Cumulative sum of TTTP scores for every year of life in the mouse hippocampus dataset. (**f**) The TTTPs for genes with the greatest expression changes prior to the TTTP (ΔE) were concentrated around the late-twenties. (**g**) Percentage of probes associated with TTTPs that up- or down-regulate prior to turning. (**h**) Percentage of probes with TTTPs which plateau or reverse after turning. (**i**) and (**j**). Age of individual mice and humans can be accurately predicted using a Support Vector Machine trained on the expression data. Individual points represent each mRNA sample.

DOI: https://doi.org/10.7554/eLife.17915.005

The following figure supplements are available for figure 2:

**Figure supplement 1.** Distribution of ALiGeT scores at 15 (a), 25 (b), 35 (c) and 55 years (d) along with examples of trajectories, their scores and expression data.

DOI: https://doi.org/10.7554/eLife.17915.006

**Figure supplement 2.** the mouse samples are divided amongst two background strains (C57Bl/6 and 129s5) and two sexes, and range from 58 to 600 days of age, whilst human samples are from both sexes and span almost 80 years.

DOI: https://doi.org/10.7554/eLife.17915.007

**Figure supplement 3.** The optimal number of probes to use for age predictions was 40 in humans (a) and 100 in mice (b).

DOI: https://doi.org/10.7554/eLife.17915.008

To characterize the features of the human TTTPs in more detail, we focussed on those genes that showed the greatest changes. As shown in *Figure 2f*, the TTTPs for genes with the greatest expression changes prior to the TTTP were concentrated around the late-twenties, reinforcing the earlier finding that this is a significant period for switching in the trajectories. Next, we separated genes into those with upward or downward trajectories prior to the turning point: overall there were similar numbers in each category, although there was a skew toward upward inflecting genes being more common in young subjects (<25 years) and downward inflecting genes more common in older subjects (>40 years) (*Figure 2g*). Finally, we examined the direction of the trajectories after the TTTP by dividing genes into those that established a stable plateau (Post-Turn Plateau) or reversed their direction (Post-Turn Reversal) (*Figure 2h*). The vast majority of probes had plateaued by 30 years of age. While the exact ages at which TTTPs occur is sensitive to both the regression method and the dataset used, it is clear that the TTTP-peak reveals a major molecular reorganisation in young adults towards the end of development. Together these findings show that young adulthood is a crucial time for switching brain gene expression and establishing the set points of most genes for later life.

Even though there are major changes during brain maturation in young adults, the complex trajectories were found throughout the lifespan, suggesting they could be used to predict the biological age of the brain. To test this, we used radial basis support vector machines and demonstrated that classifiers trained on partitioned subsets of the gene expression data (training sets) predicted age in the test sets with an accuracy (defined as mean $|Age_{Actual}-Age_{Predicted}|$) in humans of 5.5 years and 28 days in mice. Remarkably, they showed accurate age predictions across the entire range of ages in both species (human, $R^2 = 0.88$, mice, $R^2 = 0.94$) (*Figure 2i,j*) using only 40 probes in humans and 100 in mice (*Figure 2—figure supplement 3*). Thus, TTTPs and trajectories are highly characteristic features defining brain age across the lifespan in mice and humans. This indicates a 'genetic lifespan calendar' of transcriptome events is a conserved feature of mammals.

## Cell type changes throughout the lifespan

To ascertain the biological processes affected by the TTTPs, we first sought to identify the cell types affected at each age. We asked if the TTTPs were enriched in the transcriptome of specific cell types using brain single-cell RNA-seq data and the Expression Weighted Cell-type Enrichment (EWCE) method (*Skene and Grant, 2016*). TTTPs were binned into approximately equal sized age-groups (similar to the DeGeT method) and then tested for cellular enrichments (*Figure 3a*). We assumed that different biological processes could be associated with up/down-regulated genes and so performed enrichment analyses for each direction separately. To ensure the findings were robust, the analysis was performed on both the Braincloud (*Colantuoni et al., 2011*) dataset and an independent human prefrontal cortex transcriptome dataset (*Somel et al., 2009*) (see Materials and methods). Significant enrichments were found for each cell-type tested and these were strongly correlated between the two datasets (*Figure 3a*, median correlation of enrichments for each cell-type and direction with at least one significant change between the two datasets was 0.37). The majority of significant enrichments were found amongst the Upward (*Figure 1a*) gene sets relative to the Downward trajectory gene sets.

A striking sequence of events was observed where each cell type was regulated within distinct age-windows (*Figure 3a* and summarised in Figure 7). Early postnatal life was marked by TTTPs in endothelial cells ($p^{Bonferroni}_{braincloud}$=0.025, $p^{Bonferonni}_{somel}$<0.00001 where 'Bonferroni' indicates a Bonferroni-adjusted p-value) and oligodendrocytes ($p^{Bonferroni}_{somel}$=0.015); followed by microglial genes ($p^{Bonferroni}_{braincloud}$<0.00001, $p^{Bonferroni}_{somel}$<0.00001) throughout adolescence; then pyramidal neuron genes ($p^{Bonferroni}_{braincloud}$=0.0216, $p^{Bonferroni}_{somel}$<0.00001) in early adulthood (corresponding to the peak in turning points); then interneuron ($p^{Bonferroni}_{somel}$<0.00001) and oligodendrocyte genes through the late twenties and thirties ($p^{Bonferroni}_{braincloud}$<0.00001, $p^{Bonferroni}_{somel}$<0.00001). Astrocytes were found to have two periods of enrichment, the first during early adulthood ($p^{Bonferroni}_{somel}$=0.0028) and a second very late in life ($p^{Bonferroni}_{braincloud}$=0.00675, $p^{Bonferroni}_{somel}$<0.00001). These enrichments reveal the sequential maturation of cellular processes in the brain across the human lifespan. Moreover, most of these changes occurred prior to 35 years of age with prominent neuronal changes in young adults.

We also performed this analysis on the mouse hippocampal dataset (*Figure 3b*). As in the human neocortex datasets we found an early adulthood enrichment for upward-turns in pyramidal neuron

genes ($p_{mouse}^{Bonferroni}$=0.008), followed by a later enrichment for downward-turns in genes associated with both pyramidal and interneurons ($p_{mouse}^{Bonferroni}$=0.006 and, $p_{mouse}^{Bonferroni}$=0.0086 respectively). Unlike in the human datasets, upward-turns in interneuron genes were found to be enriched at the same ages as those in pyramidal neurons ($p_{mouse}^{Bonferroni}$=0.025). This may be because the earliest samples in the mouse ageing dataset were 56 days whereas the human datasets included fetal samples: correspondingly the early life enrichments for upward-turns in microglia and endothelial cells were not seen in mice. A later enrichment for upward-turns was seen in endothelial cells in mice ($p_{mouse}^{Bonferroni}$=0.048). These findings show that the enrichment of neuronal genes in the early adult peak is conserved across species and brain regions.

## Synaptic mechanisms in young adults

The young adult peak in TTTPs is a prominent landmark and we next sought to identify the key molecular mechanisms involved. We used ALiGeT to identify the top 10% of genes in both human and mouse in the year in which the largest number of TTTPs occurred (and refer to this as the Peak Gene Turning (PeGeT) score) (*Supplementary files 1a and 1b*). The relevant biological processes in the genes with high PeGeT scores, conserved between mouse and human (*Supplementary file 1c*), were first examined for enrichment in Gene Ontology terms: this revealed their role in *synaptic transmission* ($p^{FDR-adj}$=0.00024 where 'FDR-adj' indicates the p-value was adjusted for False Discovery Rate using the Benjamini-Hochberg method). Mammalian Phenotype Ontology annotations indicated their roles in *behavioural* (p=7.5×10$^{-05}$) and *nervous system* (p=0.0002) phenotypes. To probe the synaptic role in more detail, we examined genes coding for proteins in the human postsynaptic density (hPSD) (*Bayés et al., 2011*) and the postsynaptic PSD95 supercomplexes (*Husi et al., 2000*; *Frank et al., 2017*; *Fernández et al., 2009b*; *Frank et al., 2016b*), which are crucial in controlling synaptic plasticity and behaviour: both sets showed significantly higher PeGeT scores than expected by chance (p<0.0009 and p=0.0019 respectively, *Figure 4a,b*, *Supplementary file 1d*).

To establish the age window when synaptic genes showed TTTPs, we used a bootstrapping approach to test whether synaptic genes have higher ALiGeT scores in particular years in the Braincloud dataset. Each year between 22 and 33 years of age showed a significant increase in synapse-associated ALiGeT scores and we refer to this as the ALiGeT$_{hPSD}$ window. A replication study (Somel) using a smaller human transcriptome dataset (*Somel et al., 2009*) gave an overlapping estimate for the synaptic ALiGeT$_{hPSD}$ window at 17–22 years (*Figure 4c*). We next applied the DeGeT method and found all five consecutive age sets from 24 to 36 years were significantly enriched in hPSD genes (*Figure 4d*). This result was confirmed in two independent human frontal cortex datasets (*Somel et al., 2009*; *Kang et al., 2011*) (see *Figure 4—figure supplement 1*). Similarly, we identified the ALiGeT$_{mPSD}$ window in mice at 126 to 151 days (*Figure 4e*). These data show that transcripts encoding postsynaptic proteins were significantly enriched in TTTPs during early adulthood in all four datasets tested, spanning two species and two brain regions.

Since these transcriptome results suggest that specific changes occur in the composition of components of the synapse proteome, we performed relative quantitation by mass spectrometry on 38 forebrain synaptosome samples between 1–5 months in mice. A total of 900 proteins were detected, of which 99 were found to show significant changes with age (*Supplementary file 1e*, $p^{FDR-adj}$<0.005). We explored which functional classes of synapse proteins were affected by ageing during this period, and found that ion channel proteins and receptors showed increased likelihood of being differentially expressed ($p^{FDR-adj}$=0.00034, expected = 7, actual = 17, *Figure 4f*). Amongst the affected channels were the majority of $Ca^{2+}$- and $Na^+/K^+$-ATPases detected in our dataset, including three of the subunits of Atp2b (often referred to as PMCA). This finding recapitulates and extends the previous reports that aged rodents have decreased $Ca^{2+}$- and $Na^+/K^+$- ATPase activity (*Zaidi et al., 1998*; *Tanaka and Ando, 1990*). These proteomic findings support the transcriptome findings that changes in synapse proteome composition occurs within the young adult age window.

## Lifespan trajectories and schizophrenia

Prompted by previous results showing that postsynaptic proteins have been linked to the genetic susceptibility of schizophrenia (*Fromer et al., 2014*; *Fernández et al., 2009a*; *Nithianantharajah et al., 2013*; *Kirov et al., 2012*; *Purcell et al., 2014*), we hypothesized that TTTPs may be relevant to the age of onset of schizophrenia. For these analyses we examined

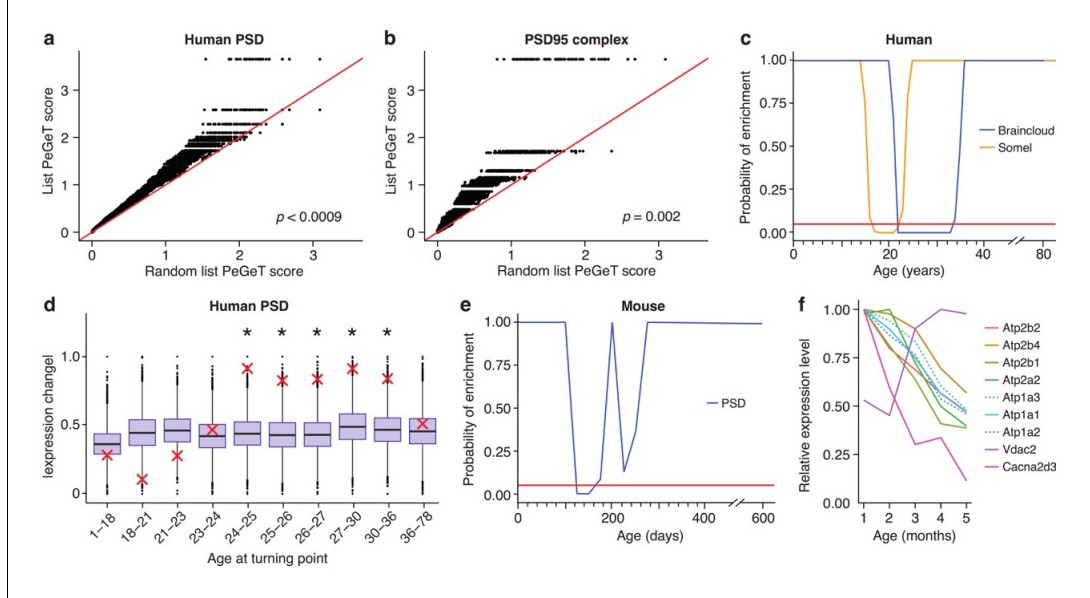

**Figure 4.** Post-synaptic density (PSD) genes are associated with TTTPs during windows in adolescence and early adulthood. (**a**) PeGeT scores for human PSD (hPSD) genes compared with scores from 100 randomly sampled gene lists. For each of the 100 random lists, the i[th] largest score is plotted against the i[th] largest hPSD score. If the distribution of PeGeT scores within the PSD gene list was normal, then an equal number of random scores would be expected on either side of the red line. P value from bootstrapping of 10,000 random lists is shown. (**b**) PeGeT scores for PSD95 supercomplex genes analysed as in (**a**). (**c**) Human PSD genes show a window of increased ALiGeT scores during young adulthood in two human prefrontal cortex datasets (Braincloud, blue; Somel, yellow). Beneath the red horizontal line marks the point of significance (Bonferroni corrected p=0.05). (**d**) DeGeT scores were significantly increased in hPSD genes in five consecutive age sets spanning 24–36 years. Boxplots show the mean bootstrapped scores (10,000 random lists) and the red cross marks the score of the hPSD list for that age. *, Bonferroni corrected p<0.05. (**e**) Human PSD genes show a window (126 to 151 days) of increased ALiGeT scores during young adulthood in a mouse hippocampal dataset. Beneath the red horizontal line marks the point of significance (Bonferroni corrected p=0.05). (**f**) The mouse synaptic proteome shows extensive changes during early adulthood, with particularly strong impact on ion channels. Expression profiles for a subset of the channels and receptors found to be differentially expressed with age. Values shown are the mean for each month, divided by the maximum mean expression value for that protein across all five months.
DOI: https://doi.org/10.7554/eLife.17915.010

The following figure supplement is available for figure 4:

**Figure supplement 1.** DeGeT enrichment for the human post-synaptic density gene set was repeated in both the (**a**) Somel and (**b**) BrainSpan datasets.
DOI: https://doi.org/10.7554/eLife.17915.011

multiple sets of publically available genetic data including de novo and GWAS data (see Materials and methods 'Genes Lists', *Supplementary file 1f*) and applied multiple analytical approaches.

As a first step, we applied the same bootstrapping approach used earlier for the PSD genes on the pooled set of genes which have been associated with Schizophrenia using either GWAS or de novo approaches. Strikingly, this analysis showed the TTTPs in susceptibility genes predicted the known age windows for the onset of schizophrenia (*Figure 5*). The ALiGeT$_{scz}$ enrichment window spanned 22–26 years (*Figure 5a*) corresponding to the clinically reported age of onset defined by the first episode of psychotic symptoms and window of maximum vulnerability (*Kessler et al., 2007*; *Häfner et al., 1993*). Since males are reported to have an earlier disease onset than females (*Kessler et al., 2007*), we tested males and females separately and found the ALiGeT$_{scz}$ enrichment window was significantly earlier in males than females (males 20–26 years, females 26–28 years; wilcoxon p=4.7×10$^{-14}$, *Figure 5b*).

To validate these results, we performed a series of technical control studies and biological replications. First, we used the DeGeT scoring system that showed the enrichment in schizophrenia peaked at 24–26 years (*Figure 5c*). Secondly, we performed the DeGeT enrichment test in the two independent human frontal cortex datasets (*Somel et al., 2009*; *Kang et al., 2011*) and confirmed that schizophrenia was significantly enriched (confirming the Braincloud result) (*Figure 5—figure supplement 1*). Thirdly, to confirm the results were not specific to a particular spline regression method, we demonstrate that enrichment window for schizophrenia was found using two alternate

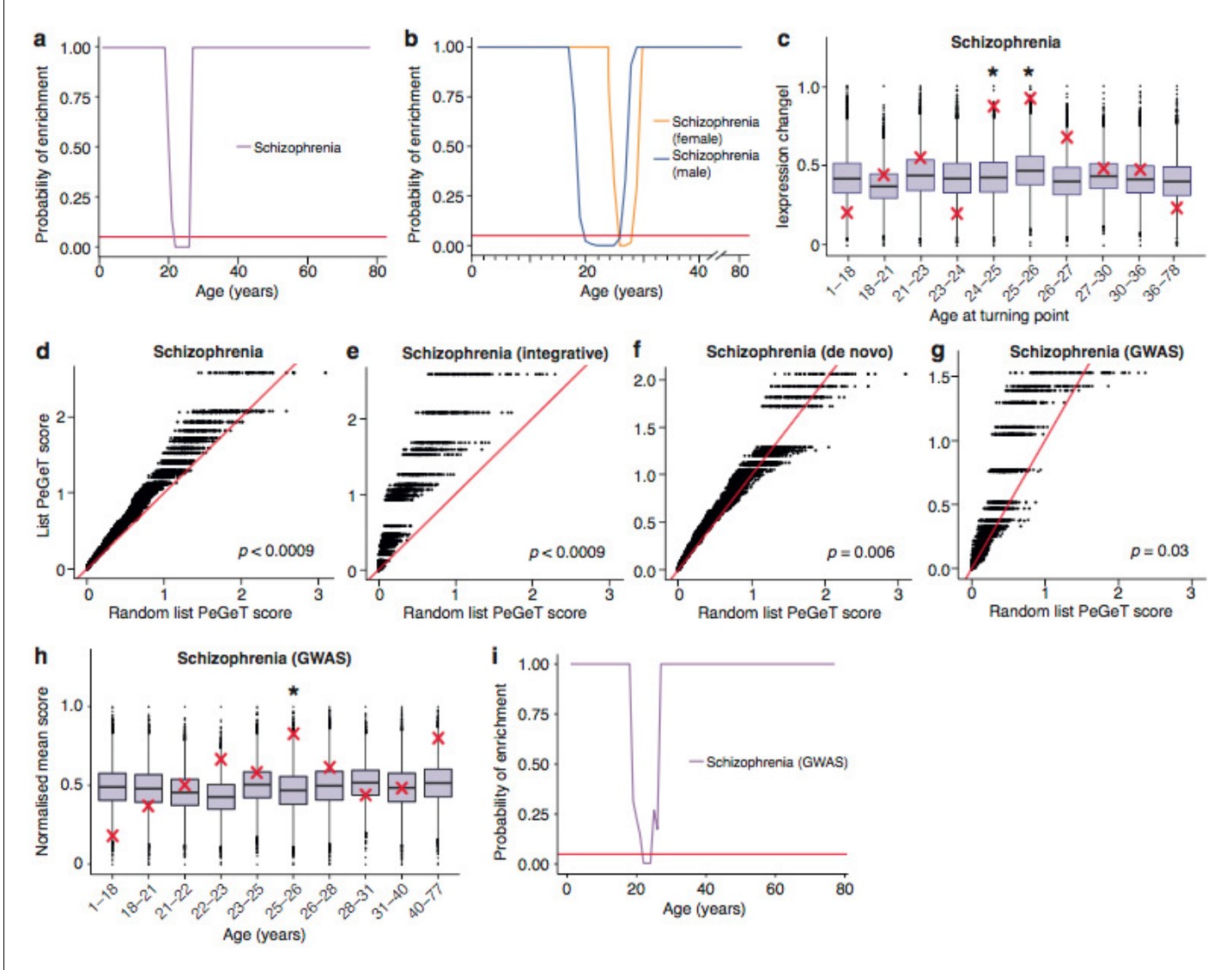

**Figure 5.** Schizophrenia susceptibility associated genes are associated with TTTPs during windows in adolescence and early adulthood. (a) Schizophrenia susceptibility genes show a window (22–26 years) of increased ALiGeT scores during young adulthood in the Braincloud dataset. Beneath the red horizontal line marks the point of significance (Bonferroni corrected p=0.05). (b) Schizophrenia susceptibility genes show a later window of increased ALiGeT scores in females (26–28 years) than in males (20–26 years). (c) DeGeT scores were significantly increased in Schizophrenia susceptibility genes in two consecutive age sets spanning 24–26 years. Boxplots show the mean bootstrapped scores (10,000 random lists) and the red cross marks the score of the schizophrenia list for that age. *, Bonferroni corrected p<0.05. (d) PeGeT scores for the schizophrenia susceptibility genes (pooled from de novo, GWAS and integrative studies) compared with scores from 100 randomly sampled gene lists. For each of the 100 random lists, the i[th] largest score is plotted against the i[th] largest disease gene score. If the distribution of PeGeT scores within the disease gene list was normal, then an equal number of random scores would be expected on either side of the red line. P value from bootstrapping of 10,000 random lists is shown. (e) PeGeT scores for the schizophrenia genes from the integrative study analysed as in (g). (f) PeGeT scores for the schizophrenia genes from the de novo study analysed as in (g). (g) PeGeT scores for the schizophrenia genes from the GWAS study analysed as in (g). (h) DeGeT scores at 25—26 years are enriched for schizophrenia heritability (calculated with GWAS summary statistics instead of associated gene set). Bootstrapping was performed by shuffling gene level association z-scores which had been calculated using MAGMA. Boxplots show the mean bootstrapped scores and the red cross marks the score with the unshuffled association scores for that age. *, Bonferroni corrected p<0.05. (i) ALiGeT scores are enriched for schizophrenia heritability (calculated with GWAS summary statistics) in the Braincloud dataset between 22 and 24 years of age. The red horizontal line marks the point of significance (Bonferroni corrected p=0.05).

DOI: https://doi.org/10.7554/eLife.17915.012

The following figure supplements are available for figure 5:

**Figure supplement 1.** DeGeT enrichment for the schizophrenia associated genes in the two additional human frontal cortex datasets.

DOI: https://doi.org/10.7554/eLife.17915.013

*Figure 5 continued on next page*

*Figure 5 continued*

**Figure supplement 2.** Early adult peak of turning points as well as significant windows for disease and synapse ALiGeT increases are robust against changes in the model used to fit the data.

DOI: https://doi.org/10.7554/eLife.17915.014

**Figure supplement 3.** Down-sampling sensitivity analysis indicates that DeGeT enrichments are stronger in earlier adulthood for schizophrenia associated genes than for the hPSD.

DOI: https://doi.org/10.7554/eLife.17915.015

**Figure supplement 4.** Enrichment of Schizophrenia associated genes does not occur as a side effect of turning points in genes with long transcript lengths or GC-content.

DOI: https://doi.org/10.7554/eLife.17915.016

**Figure supplement 5.** :The ALiGeT scoring function had a single fixed parameter, which controls the extent to which genes which turn proximally to the target year receive high scores: higher values for the parameter result in increasingly rapid decay.

DOI: https://doi.org/10.7554/eLife.17915.017

**Figure supplement 6.** Replication of results with all fetal samples are dropped from the Braincloud dataset.

DOI: https://doi.org/10.7554/eLife.17915.018

**Figure supplement 7.** DeGeT enrichment for the additional de novo gene schizophrenia gene set from the Gulsuner et al paper.

DOI: https://doi.org/10.7554/eLife.17915.019

**Figure supplement 8.** Schizophrenia heritability enrichments (calculated with GWAS summary statistics instead of associated gene set) are confirmed in the Somel dataset.

DOI: https://doi.org/10.7554/eLife.17915.020

approaches to model fitting (*Figure 5—figure supplement 2*). While validating the occurrence of the disease enrichments, this analysis also revealed that the exact age at which the turning points/ windows of enrichments occur depends on the regression model used (*Figure 5—figure supplement 2*). Fourth, we performed a down-sampling based sensitivity analysis to determine how the size of the gene set influences the enrichment: this indicated that the DeGeT enrichments are stronger for schizophrenia than for the hPSD (using subsets of 650 genes, 95% of schizophrenia subsets were significant at 24—25 years but only 50% of hPSD subsets, *Figure 5—figure supplement 3*). Fifth, we performed a variation of the bootstrapping analysis which accounts for transcript length and GC content (both of which are known to affect the rate of de novo mutations) and found no effect on the significance of the results (*Figure 5—figure supplement 4*). Sixth, we confirmed that varying the parameter in the ALiGeT scoring function, which controls the rate of decay with temporal distance, did not adversely affect the results (*Figure 5—figure supplement 5*). Finally, we dropped all fetal samples from the analysis, recalculated the TTTPs and confirmed that the major peak of TTTPs occurs in the early twenties, that it is delayed in females, that the later DEGET windows are enriched for PSD genes, and that schizophrenia shows discrete and significant enrichments using DEGET (*Figure 5—figure supplement 6*).

To further validate our findings, we next examined different genetic datasets that have been used to identify schizophrenia susceptibility genes. The ALiGeT$_{scz}$ analysis results described above used a combined schizophrenia gene set from three orthogonal genome-wide methods: (1) an integrative analysis of Genome Wide Association Studies (GWAS), expression analysis, copy number variants (CNV) and mouse models (*Ayalew et al., 2012*); (2) combined results of three exome-sequencing studies (*Fromer et al., 2014*; *Xu et al., 2012*; *Girard et al., 2011*), and (3) the most recent GWAS results from the Schizophrenia Working Group of the Psychiatric Disease Consortium (*Schizophrenia Working Group of the Psychiatric Genomics Consortium, 2014*) (*Supplementary file 1f*). We therefore separately tested sets of susceptibility genes discovered with all three methods: all showed significantly increased PeGeT scores (Integrative Analysis, $p^{FDR-adj}<0.0009$, de novo, $p^{FDR-adj}=0.0057$, GWAS, $p^{FDR-adj}=0.027$, *Figure 5d–g*). Interestingly, the stronger enrichment seen with de novo mutations may reflect that GWAS detects common variants that are assumed to have lower penetrance (*Schizophrenia Working Group of the Psychiatric Genomics Consortium, 2014*). We also validated the results using an additional set of de novo mutations (*Gulsuner et al., 2013*): this also confirmed the adult enrichments, either when analysed independently or when combined with the combined sets described above (*Figure 5—figure supplement 7*).

Much of the heritability for schizophrenia is associated with SNPs that have not reached genome-wide significance with current sample sizes (*Loh et al., 2015*) (and were not included in our analysis thus far) and the sample sizes for de novo studies are too small to determine whether any genes found are significantly associated with disease status. We therefore adapted our methods to include a greater fraction of the SNPs associated with schizophrenia heritability by using association statistics from all SNPs regardless of whether they are genome wide significant (as defined in GWAS summary statistics files) and explicitly modelling linkage disequilibrium (based on results of the 1000 genomes project), such that disease association scores can be ascertained for each gene. The ALiGeT and DEGET approaches were extended to directly utilise the gene association scores generated by MAGMA (Multi-marker Analysis of GenoMic Annotation) (*de Leeuw et al., 2015*) based on schizophrenia GWAS summary statistics. Using this approach, schizophrenia showed a DEGET enrichment at 26 years in the Braincloud dataset ($p^{Bonferroni}$=0.03135, *Figure 5h*) and at 15—16 years in the Somel dataset ($p^{Bonferroni}$=0.0115, *Figure 5—figure supplement 8a*) and corresponding enrichment windows were found using ALiGeT (*Figure 5i*, *Figure 5—figure supplement 8b*).

Finally, we performed two additional sets of analyses where we examined mouse proteome datasets and single-cell transcriptome data. Consistent with the transcriptome results, the mouse synapse proteomic dataset showed that schizophrenia-associated proteins were enriched in the synapse proteins that changed prior to the 5 month peak (p=0.018, expected = 6, actual = 11), and 91% of those that change were found to be down-regulated. Next, we examined the role of specific cell types in schizophrenia by restricting the ALiGeT analysis to the subsets of genes defined by single cell transcriptome data (see Materials and methods) (*Zeisel et al., 2015*). Schizophrenia associated genes showed a significant ($p^{Bonferroni}$<0.05) window of enrichment in neuronal genes (*Figure 6a*, *Figure 6—figure supplement 1*). We were however unable to confirm this result using the summary statistics based method (*Figure 6b*). We then examined the types of trajectories associated with schizophrenia and found genes were predominantly down-regulated prior to the TTTP (*Figure 6c*). This result was confirmed using the summary statistics based method (*Figure 6d*).

Together these analyses show robust replication across multiple datasets, including different types of genetic variants, transcriptomes and proteomes, and several complementary analytical approaches. These data strongly suggest that the window of onset and sex difference in schizophrenia is timed by the regulation of susceptibility genes in young adults.

## Synapse proteome and adult-onset neurological disorders

In the mouse synapse proteome dataset, we noticed several proteins that cause adult onset monogenic disorders. We used the Human Phenotype Ontology age-of-onset annotations to identify these neurological disorders (excluding neoplasms, and peripheral/autonomic disorders) and from a total of 39 genes, six were detected in our samples, and we confirmed these were significantly enriched amongst the synapse proteins whose expression levels changed during maturation (p=0.00008, expected = 1, actual = 4, *Figure 6—figure supplement 2*, *Supplementary file 1e*). The affected proteins/disorders included Atp2a2 (Darier's disease), Eef2 and Itpr1 (Spinocerebellar ataxia), Atp1a3 (Dystonia Parkinsonism). No equivalent enrichment was found for congenital/neonatal onset disorders (p=0.45, expected = 4, actual = 4). Further inspection (because HPO annotations were incomplete) found three additional, adult-onset haploinsufficiency disorders encoding by genes showing a 50–80% reduction in the maturing mouse synapse proteome (*Figure 6—figure supplement 2*): Vcp (Frontotemporal Dementia) (fold change, FC = 0.48; p=0.002); Atl1 (hereditary spastic paraplegia) (FC = 0.19, p=0.00004); Dmxl2, (Polyendocrine-polyneuropathy syndrome) (fold change: 0.28, p=0.00004). These age-dependent reductions in levels of haploinsufficient disease genes is consistent with the model that their respective lifespan trajectories are relevant to their age of onset.

## Prefrontal cortex early adult turning points are not associated with other polygenic brain disorders

The possibility exists that the lifespan trajectories are also relevant to the age of onset of other polygenic diseases with onset at different ages. To address this we compiled gene lists for six other major brain disorders with different age-windows of onset: onset during infancy (autism and intellectual disabilities); early adulthood (multiple sclerosis); and late adulthood (Amyotrophic Lateral

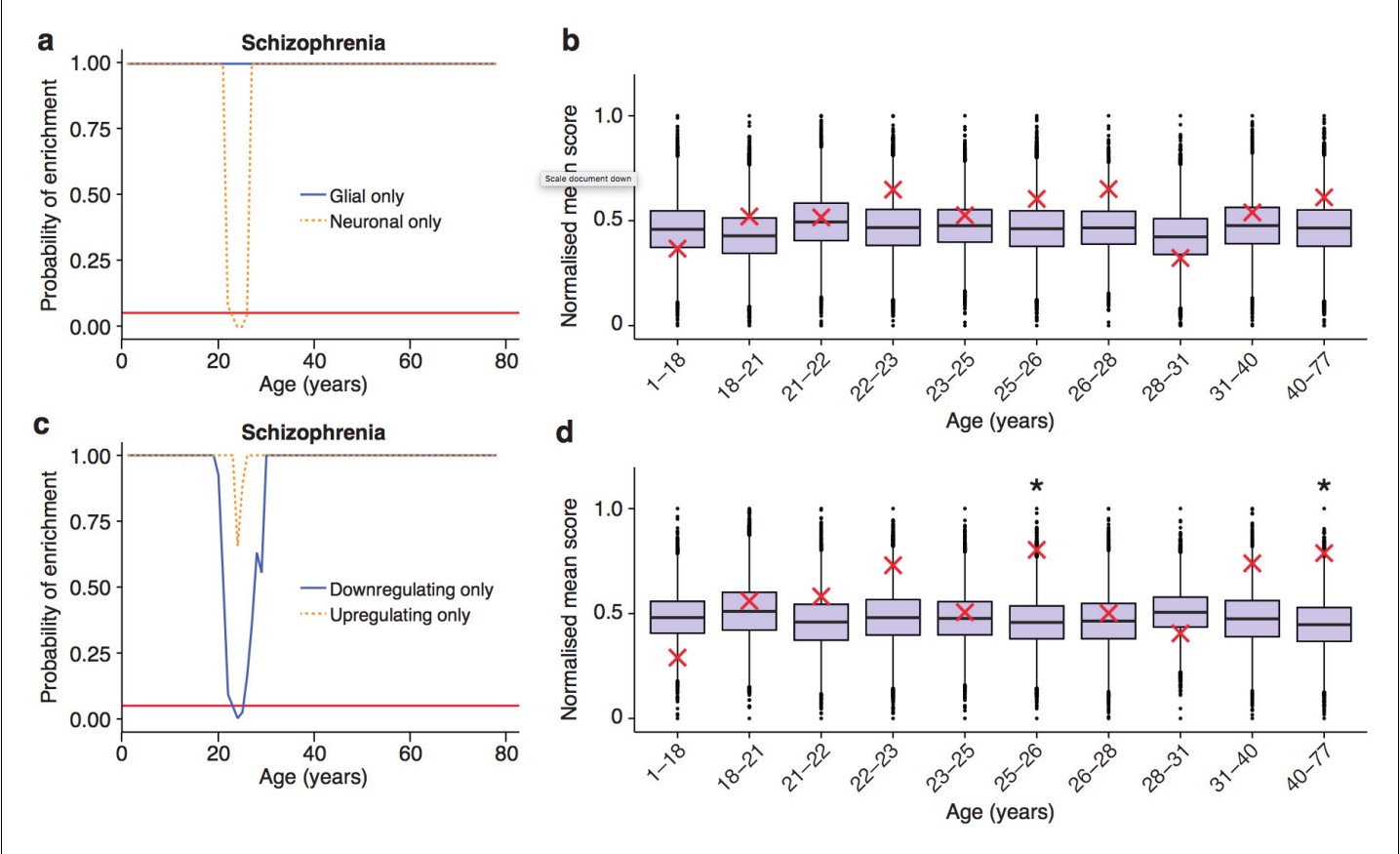

**Figure 6.** Trajectories of schizophrenia associated genes show further distinctions in their cell-type and trajectories. (a) Schizophrenia susceptibility genes show a window of increased ALiGeT scores when the analysis is restricted to the 5000 most neuron specific genes, but not the 5000 most glial specific genes. The red horizontal line marks the point of significance (Bonferroni corrected p=0.05). (b) DeGeT scores of the 5000 most neuron enriched genes do not confirm a significant enrichment in schizophrenia heritability calculated with GWAS summary statistics. Bootstrapping was performed by shuffling gene level association z-scores which had been calculated using MAGMA. Boxplots show the mean bootstrapped scores and the red cross marks the score with the unshuffled association scores for that age. *, Bonferroni corrected p<0.05. (c) Schizophrenia susceptibility genes show a window of increased ALiGeT scores, when the analysis is restricted to genes that down-regulate prior to their TTTP, but not with genes that up-regulate prior to their TTTP. (d) DeGeT scores of genes which down-regulate prior to their TTTP are significantly enriched for Schizophrenia heritability calculated with GWAS summary statistics at 26—27 and 40–77 years.

DOI: https://doi.org/10.7554/eLife.17915.021

The following figure supplements are available for figure 6:

**Figure supplement 1.** Enrichments for schizophrenia susceptibility genes are specific to particular cell types.

DOI: https://doi.org/10.7554/eLife.17915.022

**Figure supplement 2.** The synaptic proteome was found to show extensive changes during early adulthood, with particularly strong impact on proteins associated with adult-onset mental disorders.

DOI: https://doi.org/10.7554/eLife.17915.023

**Figure supplement 3.** No cognitive disorders other than Schizophrenia showed larger PeGeT scores than expected by chance.

DOI: https://doi.org/10.7554/eLife.17915.024

Sclerosis, Parkinson's and Alzheimer's) (corresponding gene sets shown in **Supplementary file 1f**). We applied the same bootstrapping approach used earlier for the PSD genes and none of the disorders showed significant results (**Figure 6—figure supplement 3**). Because these gene lists are not comparable (different size, population sample size, obtained using different technical approaches etc.) the relative importance of the genetic calendar to schizophrenia cannot be directly compared with these disorders (see Discussion). In addition, because the transcriptome data is from prefrontal cortex and the primary pathology of several of these diseases is in other parts of the nervous system it cannot be assumed that the transcriptome trajectories in one part of the brain are the same as in

others. Hence, the lack of any detectable age window does not preclude a role for gene regulation in the onset of these diseases.

## Discussion

Understanding gene expression in the human brain during the phases of childhood, adolescence, young adulthood, middle and old age, is a fundamentally important area of biology with medical significance. We focussed on identifying age-dependent gene regulatory events that were detected when the trajectory in the level of gene expression changed. Studying the transcriptome trajectories across the lifespan of the human neocortex and mouse hippocampus showed that TTTPs occurred at all ages. Moreover, because these events were a defining characteristic of every age, we found that actual age could be predicted by examination of an RNA sample from mouse and human brain tissue. These findings indicate there is a 'genetic lifespan calendar' that sets the date for gene expression changes in both species. This conclusion complements previous epigenomic studies that show DNA methylation correlates with chronological age (*Horvath, 2013*). The most striking and unexpected feature was the peak of TTTPs in young adult humans at 26 years of age and 5 months of age in mice. In both species, this peak was delayed in females and involved similar sets of genes. Moreover, in both species there was a similar sequential pattern of cell-type specific changes across the lifespan. Thus, we conclude that mammals with greatly differing lifespans share a conserved genomic program regulating the sequence of cellular and synaptic changes throughout the lifespan.

We discovered that the young adult brain undergoes a dramatic reorganisation of gene expression, as revealed by the TTTP-peak around 26 years, and this reorganisation is largely completed by 40 years of age when ninety percent of trajectories have plateaued. These findings correspond well with anatomical data showing that development and myelination in the frontal cortex is over (*Lebel et al., 2012*) by this age. The TTTP-peak was enriched in neuronal and synaptic genes expressed in pyramidal and interneurons including those encoding PSD proteins and the 1.5 MDa PSD95 supercomplexes (*Figure 7*). This indicates that young adults undergo a major reorganisation of their synapse proteomes, which was supported by proteomics results in mice. The level of expression of many PSD and PSD95 supercomplex proteins are known to be important for many innate and learned behaviours and we also found that the TTTP-peak was enriched in behaviourally important genes. Thus the genetic calendar can modify synapse proteome composition and potentially shape the behavioural repertoire across the lifespan.

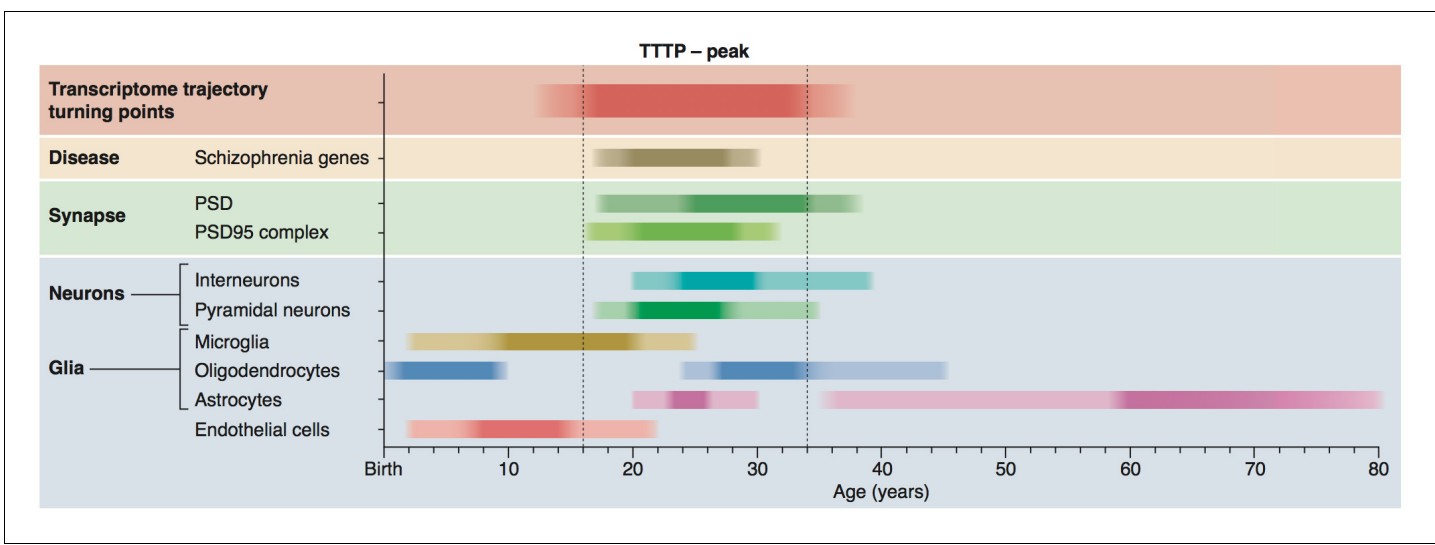

**Figure 7.** Summary of age windows for synaptic and cellular processes and diseases across the human lifespan. The intensity of the shaded boxes indicates the enrichment in relevant genes. The TTTP-peak in young adults (vertical dotted lines show approximate boundary) coincides with synaptic and neuronal changes and schizophrenia genetic susceptibility. Changes in glial cells overlap with the TTTP-peak and occur in distinct windows across the lifespan.

DOI: https://doi.org/10.7554/eLife.17915.025

The TTTP windows in non-neuronal cells also corresponded well with known changes in these cells. The early age-window in endothelial cells corresponds to the expansion and maturation of the brain's vasculature (*Caley and Maxwell, 1970*; *Engelhardt and Liebner, 2014*; *Azmitia et al., 2016*) and post-natal shift in the transcriptome of endothelial cells (*Daneman et al., 2010*). There was also an enrichment in oligodendrocyte genes early in life, potentially corresponding to downregulation of genes specific to oligodendrocyte precursor cells (*He et al., 2009*). The next phase, which continues through the early teenage years, involves strong upregulation of microglial genes and coincides with synaptic pruning (*Schafer et al., 2012*) and preceded the young adult window of synaptic and neuronal reorganisation.

## Transcriptome trajectories and schizophrenia

To date, there has not been a satisfactory mechanism or model that accounts for the following five central features of schizophrenia. First, the genetic susceptibility: the mechanism needs to account for the diverse sets of genes and the diverse types of mutations. Second, the age of onset: the mechanism needs to account for the age-window during which first-episode psychosis occurs, the heritability of this onset, as well as the earlier presence of prodromal cognitive symptoms (*Koutsouleris et al., 2012*). Third, the sex difference: females have a later onset than males. Fourth, the cell biological mechanisms: there needs to be a common subcellular mechanism that incorporates the diverse classes of disease-relevant proteins, which include channels, receptors, synaptic adhesion proteins, scaffold proteins and signalling molecules. Fifth, the cognitive deficits: the molecular and cellular mechanisms need to play a key role in the relevant cognitive processes. A model that satisfies these criteria would also be expected to have explanatory power for other characteristics of schizophrenia.

Our findings meet all five criteria and we propose the following model. A genetic program orchestrating transcriptome trajectories causes reorganisation of expression of synapse proteins in young adults. Mutations in these genes are functionally exposed in young adults because the reorganisation of expression produces inappropriate synapse signalling properties that result in abnormal behaviour. We refer to this as the genetic calendar model of schizophrenia. This model posits that schizophrenia is a genetic disorder targeting the mechanisms of brain aging during the young adulthood period of the lifespan. Our model offers a mechanistic explanation for the onset of first-episode psychosis and is consistent with prodromal cognitive impairments and the persistence of schizophrenia beyond the young adult years (when many genes reach their plateau). Thus, the genetic calendar model can explain the onset and progression of schizophrenia.

In addition to the robust association of schizophrenia genes with the TTTP-peak which was replicated across datasets including multiple types of genetic variants, transcriptomes and proteomes, and several complementary analytical approaches, our study identified specific molecules that further strengthens the mechanistic link between synaptic mechanisms and schizophrenia. PSD95 supercomplex proteins are enriched in schizophrenia genes (*Fromer et al., 2014*; *Fernández et al., 2009a*; *Singh et al., 2017*; *Kirov et al., 2012*; *Purcell et al., 2014*; *Grant et al., 2005*). Amongst the schizophrenia-susceptibility genes with the most prominent TTTPs were those with established synaptic functions, including Rgs4, Snap25, Kalrn, Htr2a and Nrg1. The expression level of each of these genes has previously been shown to either influence, or be influenced by psychiatric symptoms (*Etain et al., 2010*; *Guillozet-Bongaarts et al., 2014*; *Yin et al., 2013*; *Hill et al., 2006*). Furthermore, altered expression of Snap25, Htr2a and Nrg1 are noted to associate with earlier age of onset (*Etain et al., 2010*; *Weickert et al., 2012*; *Abdolmaleky et al., 2011*).

## Neurological and early onset disorders

We found evidence that several mendelian neurological disorders with adolescent and young adult onset also involved proteins that were down-regulated in the TTTP-peak. Heterozygous mutations in *Atp2a2* cause Darier's disease and significantly increase the risk of many psychiatric disorders including mood disorders, depression, and schizophrenia (*Gordon-Smith et al., 2010*). Mutations in *Eef2* and *Itpr1* cause spinocerebellar ataxia type 26 and 15/29, respectively. *Atp1a3* mutations cause rapid-onset dystonia Parkinsonism, which leads to Parkinson's-like symptoms appearing during early adulthood, often with concurrent emergence of psychiatric symptoms (*Brashear et al., 2012*). Vcp mutations cause a form of frontotemporal dementia with a mean age of onset in the mid-thirties

(*Watts et al., 2004*). One of the main causes of hereditary spastic paraplegia (HSP) are heterozygous mutations in Atl1, which has an age of onset ~21 years (*McCorquodale et al., 2011*). Interestingly, we found that the synaptic scaffold protein Dmxl2 (haploinsufficiency causes Polyendocrine-poly-neuropathy syndrome [*Tata et al., 2014*]) was reduced by ~70% (fold change: 0.28, p=0.00004) and studies in mice show heterozygous deletion of Dmxl2 in central neurons delayed the onset of puberty (*Tata et al., 2014*). These findings support the view that the genetic lifespan calendar reduces expression below a critical threshold in young adults and is important for multiple neurological and psychiatric diseases.

Our studies relied on human prefrontal cortex transcriptome data, which may have limited our ability to detect a role for the genetic lifespan calendar in the age-dependent onset of those diseases that are known to have primary pathology in other brain regions (e.g. Parkinson's disease). Given the evolutionary conservation between mouse hippocampus and human prefrontal cortex, we expect other human brain regions to show a calendar of transcriptome trajectories, but with different patterns and therefore different age windows of disease gene enrichments. Moreover, while our whole tissue transcriptome analysis appeared to be sensitive to neuronal changes, we expect that future single-cell transcriptome data will provide a more detailed insight into rarer cell types and potentially reveal mechanisms relevant to the age of onset of pathology in these cells. We do not expect that the age of onset of all brain diseases will be accounted for by the genetic calendar, as it will likely depend on the importance of cell autonomous processes and exogenous factors (e.g. inflammatory processes involving microglia). It is also likely that high penetrance severe mutations will show an earlier onset and are less likely to show a dependence on the TTTPs. Weaker alleles may manifest with undetectable or subtle phenotypes at early ages and be exposed at later ages by the changes in gene expression. Thus, TTTPs would not necessarily account for the age of onset of intellectual disability or autism even though some of the same genes are involved with schizophrenia.

## Role of genetic lifespan calendar in behavioural adaptation

The genetic lifespan calendar is an innate mechanism that regulates the levels of postsynaptic proteins and hence changes the physiological and behavioural properties of brain circuits. This indicates that the brain is not 'hard wired' but is continuously being modified by an evolutionary ancient and conserved genetic program. The postsynaptic proteins control innate and learned behaviours and thus the genetic calendar will modulate innate behaviours and the capacity to learn across the lifespan. The overarching biological function of this program could be to equip the animal for the challenges it faces at different ages.

If there are mutations that interrupt this calendar of events, then the organism will not respond appropriately to environmental challenges. This may be relevant to schizophrenia where exogenous factors are thought to influence onset (e.g. cannabis) or behaviour (e.g. smoking) during young adulthood. Our model which posits a fundamental role for genetic and genomic mechanisms can therefore accommodate previous models that have considered exogenous factors in disease aetiology. Interestingly, environmental triggers of psychosis, such as cannabis and other drugs of abuse, are known to act on the synaptic signalling mechanisms (*Camp et al., 2011*; *Abbas et al., 2009*) that are being reorganised in the TTTP-peak. Our model also has implications for those schizophrenia models that posit a (non-genetic) fetal insult as the predisposing factor for later onset (*Brown and Derkits, 2010*): the enrichments in schizophrenia susceptibility genes in young adults was present even when fetal samples were not included in the analysis supporting the notion that it is a disorder of postnatal brain ageing.

Our genetic lifespan calendar model may also have implications for the development of pharmaceuticals for the adolescent and young adult onset psychiatric and neurological disorders. As an alternative to the 'precision medicine' approach which directs pharmacological treatments to each susceptible genotype, we suggest that therapeutics accommodating many genotypes might lie in drugs that modify the genetic lifespan calendar. The identification of the conserved TTTP-peak between mice and humans may also assist in refining animal models of adult-onset brain disorders. Our findings open a range of new approaches for understanding brain ageing and the mental disorders afflicting young adults.

## Materials and methods

### Mice

Hippocampi from 186 mice of both sexes and two background strains (C57Bl/6 and 129s5) aged 58—600 days were used for the mouse microarray dataset (*Figure 2—figure supplement 2*). All animal experiments conformed to the British Home Office Regulations (Animal Scientific Procedures Act 1986; Project License PPL80/2,337), local ethical approval, and NIH guidelines. Animals were born and raised within the Research Support Facility at the Wellcome Trust Sanger Institute, and exposed to the same light/dark cycles and feed supply. Mice were exported to a holding facility three days prior to collection of brain samples, to allow time for stress response genes induced through movement to subside. All animals were sacrificed by cervical dislocation, and the hippocampi were dissected on ice, and snap frozen in liquid nitrogen.

### RNA preparation and microarray hybridization

Mouse brain samples were homogenised using the Kontes Cordless Pellet Pestle system. Total RNA was extracted using the Qiagen miRNeasy kit, snap frozen using liquid nitrogen and stored at −80°C. Microarray processing was performed by the Wellcome Trust Sanger Institute's Microarray facility. The Illumina TotalPrep-96 RNA Amplification kit was used for reverse transcription, amplification, and biotinylation of the RNA prior to hybridisation. The microarrays used were the Illumina MouseWG-6 v2 series. Hybridization, washing, and staining were performed according to standard Illlumina protocol. The microarrays were imaged using the Illumina BeadArray Reader. Images from the scanner were processed using the BeadStudio software. The microarray data is available to download from ArrayExpress (accession number E-MTAB-3256).

### Braincloud human prefrontal transcriptome dataset

The human samples were generated by the Braincloud project (braincloud.jhmi.edu) as described in their primary publication (*Colantuoni et al., 2011*). In brief they used post-mortem human brains from the NIMH Brain Tissue Collection, National Institute of Child Health and Human Development Brain and Tissue Bank for Developmental Disorders. RNA samples were described as being extracted from ~100 mg of tissue from BA46/9, the dorsolateral prefrontal cortex. Custom-spotted two-colour oligonucleotide microarrays constructed from a set of oligos referred to as HEEBO7 (Human Exonic Evidence Based Oligonucleotide) were used. The preprocessed/normalized Braincloud expression data was downloaded from GEO (GSE30272, data contained within the Series Matrix file). No further normalisation was done beyond that which was already performed as described in the Braincloud publication. Low intensity probes and outliers had already been removed and expression levels were adjusted for sex, ancestry and a single surrogate variable using the sva package (*Leek et al., 2012*).

### Somel human prefrontal transcriptome dataset

The 'Somel' dataset used to validate the increased PeGeT/DeGeT/ ALiGeT scores of schizophrenia genes, and EWCE enrichments, was downloaded from the GEO repository (accession GSE11512) (*Somel et al., 2009*). The dataset comprised samples taken from dorsolateral prefrontal cortex, which had been hybridised to Affymetrix Human Genome U133 +arrays. The dataset was read into R using the bioconductor '*affy*' package, and processed using RMA. Probes with detection probabilities of less than 0.05 in over ten of the arrays were dropped.

### BrainSpan human frontal transcriptome dataset

The BrainSpan RNA-Seq gene level dataset (*Kang et al., 2011*) was downloaded from the BrainSpan website (brainspan.org) on the 21st Jan 2016. The data was read into R. Data from the following four regions were retained: dorsolateral prefrontal cortex, ventrolateral prefrontal cortex, orbital frontal cortex and anterior (rostral) cingulate (medial prefrontal) cortex. Ages were converted to numerical equivilents. Any transcripts whose summed expression over all retained samples was less than 0.1 were dropped. Because the oldest samples in the BrainSpan dataset are only 40 years of age we treated trajectories which did not turn as turning in final year of life.

## Mouse hippocampal transcriptome dataset

The mouse microarray data was read into R using the Bioconductor package Lumi (*Du et al., 2008*) (RRID:SCR_012781) and re-annotated based on a published analysis (*Barbosa-Morais et al., 2010*). The *detectionCall* function from the lumi package was used to drop probes undetected by the microarrays. Probes rated as having a '*bad*' quality (according to the re-annotation package) were dropped, as were those with the coding zone given as '*Transcriptomic?*'. A variance stabilizing transformation was applied followed by quantile normalization.

## Spline fitting

Natural cubic splines with three degrees of freedom were fitted to the expression data using the R package *splines*. For the mouse expression data, age was modeled as the independent variable, with sex and background as covariates. For the human expression data, as variation attributed to the extraneous variables (sex and ancestry) had already been subtracted earlier using the *sva* package these terms were not included. The location of knots for the splines was determined by the quantiles of the data. For each species and gene, trajectories were then interpolated across the lifespan with extraneous variables (sex and ancestry/background) held constant. We refer to these interpolated spline trajectories as Brain Transcriptome Lifespan Trajectories (BTLTs).

## Detecting turning points (TTTPs)

Prior to detecting TTTPs, spline models were fitted to the data. To calculate where a TTTP occurred in a gene's expression trajectory ($A_g$), the derivative was approximated for each BTLT and the age of the TTTP was taken as the first point where $dE/dA$ =0 and the derivative changed sign, where $E$ is expression level in the spline model and $A$ is age. If multiple turning points were found in a single spline, then only the first was used. *Figure 2a–c* show the number of TTTPs found across the transcriptome within a given year/month. To determine the TTTPs for each sex, all samples from the other sex were dropped, the splines remodeled (without including sex as a covariate) and TTTPs detected afresh.

The numbers presented for the age of the TTTP peak in paragraph six are the mean age of TTTPs across all transcripts. To test for a difference in the age of TTTPs between the sexes, the Wilcoxon signed rank test was used. To test whether X-chromosome genes caused the sex difference, TTTPs in genes from that chromosome were dropped the same statistical test performed on the remaining set, and a similarly significant probably for difference was found.

To analyse the properties of trajectories with TTTPs (for generation of *Figure 2g,h*), linear models were fitted to the expression data before and after each TTTP, and tested for whether there is a significant change. A small fraction of probes which were detected as having TTTPs were not found to show significant differential expression prior to the TTTP, after multiple hypothesis correction with the Benjamini-Hochberg correction (FDR = 0.05) and these were excluded from this analysis of TTTP characteristics. We classified genes as showing *post-turn plateaus* if there were not detected as differentially expressed after the TTTP with FDR = 0.05. Those that were detected as showing significant differential expression after the TTTP, in the opposite direction to before the TTTP, are denoted as showing *post-turn reversals*. To generate *Figure 2f* we calculated the deciles of the ages at which TTTPs occur, and used these to split the data into ten groups with as closely matching sizes as possible.

## Scoring TTTPs

The Age-linked Gene Turning scores (ALiGeTs) were designed to represent two features: (1) proximity of the TTTP close to the target year, and (2) the changes in their expression level from the start of the dataset (at 14th gestational week) through to the age at which the TTTP occurred (this change is denoted as $\Delta E$). A human genes ALiGeT scores for a given age ($S_{g,T}$) can be calculated as:

$$S_{g,T} = 1.5^{-|A_g-T|} * |\Delta E|$$

Wherein
$g$ is the index of the gene of interest
$T$ is the year of age being queried
$\Delta E$ is the change in expression level prior to the TTTP

$A_g$ is the age of the first TTTP in gene $g$

For mice the equation is modified slightly to account for the difference in lifespan of the two species. To allow the ALiGeT window to have an approximately equal width across a given proportion of the mouse's, we scaled the term $|A_g - T|$ by 650/78 where 650 days represents old age for a mouse and 78 represents old age for a human. A mouse genes ALiGeT score for a given age can thus be calculated as:

$$S_{g,T} = 1.5^{-\frac{|A_g - T|}{\left(\frac{650}{78}\right)}} * |\Delta E|$$

To penalize genes which are further from the peak of TTTPs, the distance between the age of turning point ($A_g$) and the target age ($T$) is used as the exponent in an exponential expression with base 1.5. This function gives greatest emphasis to genes with D = 0, with a rapid fall over neighbouring years such that a human gene with D = 6 years have scores 9% the size of those with equivalent pre-turn changes in expression level at D = 0. The scoring function is graphically depicted in *Figure 1—figure supplement 1*. We show in *Figure 5—figure supplement 5* that the results are stable to variation in the exponent base.

The term PeGeT score is used to refer to the ALiGeT score for the year in which that species has the most TTTPs.

## DeGeT scoring

To score genes using the DeGeT method (represented in *Figure 1b* and used to generate *Figure 4d* and *Figure 5c*) a similar scoring system was used. First ages were divided into ten groups based on the number of genes which turn/inflect within that age period, such that each age set has approximately equal size number of turning genes. Because there are many more TTTPs during early adulthood than in infancy/old age, the age windows span many years at start/end of the lifespan and as few as one year in during the twenties. For age set S, which covers ages $\{x, x+1, \ldots, y-1, y\}$, the score was equal to the pre-turn expression change if the gene turns within the age window bounded by x and y. If the gene does not turn within within the age window bounded by x and y then the score for that gene is zero.

## Cellular enrichment analysis with EWCE

Cell type enrichment analysis was performed on three datasets: Somel (*Somel et al., 2009*), Braincloud (*Colantuoni et al., 2011*) and the mouse hippocampus. The EWCE ('Expression Weighted Celltype Enrichment') package from Bioconductor (RRID:SCR_006442) was used to perform the enrichment with 100,000 bootstrap replicates used for each test. Age groups were assigned based on the frequency of TTTPs across the lifespan (0—10th, 11th—20th,...,90th—100th percentile of all TTTPs). The genes which turn within each age window were sorted based on the size of pre-turn expression change. For each age window, the 10% of genes which turn within that window with largest positive (upward) and negative (downward) expression changes were assigned into two groups: we refer to these lists as the 'target' lists. A previously published single cell transcriptome (SCT) dataset containing cells from cortex and hippocampus was then loaded (*Zeisel et al., 2015*). Any genes in the target lists which were not found to be expressed in the SCT dataset were dropped. For the analysis with Braincloud and Somel, the background gene set was taken as all genes with mouse orthologs that are also found in the SCT dataset for which a spline was fitted. For the mouse hippocampus analysis, the background set contained all genes for which a spline was fitted and which was also detected in the SCT dataset. For the directional analyses, the background set thus contains genes which change in both directions. Bonferroni correction was used to adjust for multiple testing. For both directions, for each cell type within each age group (indexed by r, c and $a \in A$ respectively), we calculated the mean ($\mu_{r,c,a}$) and standard deviation ($\mu_{r,c,a}$) of the bootstrap distribution, and used this to determine the distance (in terms of standard deviations) that the target list falls from the expected mean—we refer to this value as $d_{r,c,a}$. Values were calculated separately for each dataset. The plots shown in *Figure 3* show normalised values derived from $d_{r,c,a}$ by dividing by the maximal absolute value over all age windows. As a result the maximal absolute enrichment is either 1 or −1. The values plotted on the x-axis each represent one of the age windows defined by the quantiles (specifically, the values shown are the mean of the upper and lower bounds of the

window). The age windows for Somel and Braincloud are not fully overlapped but were annotated with the central points of the Braincloud age windows to enable both datasets to be plotted against each other.

## Testing for increased turning point scores of PSD and disease gene sets

To determine whether gene sets show larger ALiGeT scores than expected, a boot strapping test was performed. Genes in the target list, which were not detected as expressed on the arrays, were dropped from the analysis. Where multiple probes target the same gene, duplicated probes with smaller PeGeT inflection scores were dropped. For each target list being tested, ten thousand random gene lists of the same length were generated. The mean ALiGeT/PeGeT score for the target list and the random gene lists was calculated, and the proportion of random lists with smaller mean scores than the target list is taken as the probability. Where multiple lists were tested for significance against PeGeT scores, these were corrected using the Benjamini-Hochberg method (FDR = 0.05). The same method was used for the DeGeT set based analysis method as was used for the ALiGeT analysis, with bootstrapping being done with 10000 samples and correction for multiple testing over age sets done with the Bonferroni method.

*Figure 4a,b*, *Figure 5d–g*, and *Figure 6—figure supplement 3* represent the results of the PeGeT bootstrapping analysis in graph form. We sought to represent the extent to which enrichment results from high scores amongst either a small number of genes, or from a broad increase in scores throughout the list. As in the bootstrapping analysis we compare the score distribution in the target (disease/synapse) list to the scores in random lists of the same length. To keep the plot tidy we only represent data from 100, rather than 10000 random lists. PeGeT scores for the target and random lists were sorted by numerical size. For each of the *n* genes in the target list, 100 dots were positioned with the y-axis determined by $i^{th}$ largest score in the target list, and the x-axis given by the $i^{th}$ largest score in each of the 100 random lists. To interpret the graph, note that if the majority of the random list scores fall above the red line, then the random lists have lower scores than the target list. It should be noted that the scales of the graph does bias the view towards the genes with largest scores.

To perform a bootstrapping analysis which controls for gene size and GC content, we obtained those values from biomart. Where multiple transcript lengths were associated with a single HGNC gene we took the mean value. The deciles of gene size and GC content were calculated over the set of expressed genes. The two sets of decile values were used to define a grid, and each gene assigned to a position within the grid based on it's transcript lengths and GC content. To run a bootstrap analysis on a particular target list, 10000 random lists were constructed with equal length to the target list. Gene *i* in each random list was selected from the same grid square as gene *i* in the target list.

## Disease gene lists

The disease gene lists are provided in *Supplementary file 1f*. The combined schizophrenia gene list (results shown in *Figure 5a–d*, *Figure 6a,b*, *Figure 5—figure supplement 1–5,7* and *Figure 6—figure supplement 1*) used genetic associations from three types of studies, which were also analysed individually: the integrative dataset contains 42 genes assembled using a translational convergent functional genomics approach (*Ayalew et al., 2012*); the de novo gene set comprised 609 genes pooled from three studies which used parent-child trios to detect de novo mutations (*Fromer et al., 2014*; *Xu et al., 2012*; *Girard et al., 2011*); the GWAS results are from the 2014 release of the Schizophrenia Working Group of the Psychiatric Genomics Consortium (*Abbas et al., 2009*). Many SNPs from the GWAS study results were associated with multiple genes (349 genes were associated with 108 loci, with numerous loci associated with over twenty genes), and these were dropped leaving 62 genes to be used in our analyses (not that this does not apply to the analyses which directly utilised the GWAS summary statistics files). The GWAS result remained significant if the alternative approach was taken and all genes associated with SNPs were used. The additional schizophrenia de novo gene set (the 'Gulsuner' set) came from exome sequencing of 'quads and trios' (patients, their parents and unaffected siblings) associated with 105 individuals with the disorder (*Gulsuner et al., 2013*). Two autism lists were used, both based on finding de novo mutations through exome sequencing: the first contained 172 mutations (*Sanders et al., 2012*), the second 358

(Iossifov et al., 2012). The 78 intellectual disability genes were discovered through de novo sequencing of family groups (de Ligt et al., 2012). The Multiple Sclerosis, Amyotrophic lateral sclerosis (Lill et al., 2011), Parkinsons (23andMe Genetic Epidemiology of Parkinson's Disease Consortium et al., 2012) and Alzheimer's (Bertram et al., 2007) lists come from the top results of the following websites: msgene.org (69 genes); alsgene.org (17 genes); pdgene.org (23 genes); alzgene. org (10 genes).

## ALiGeT enrichment windows

Gene list enrichments were evaluated across the lifespan by generating $S_{g,T}$ for each gene, for each value of $T$ between 1 through 78 ($S_{g,T}$ defined above). To test for disease enrichments the boot strapping method described above was applied. Enrichment of the gene sets were calculated at each age. The Bonferroni method was used to correct for the testing of the hypothesis at each of the 78 years of age. When running the ALiGeT for the eight diseases Bonferroni correction was applied across all the diseases as well as over each year of age. To obtain the sex specific datasets, the enrichments were calculated separately on the male subset of the data, and then again on the female subset. To test for sex differences in TTTP ages specific to disease gene sets, the age of TTTPs associated with those genes in the male and female data subsets were compared using the Wilcoxon signed-rank test.

To control for transcript length, the maximum transcript length was determined for each gene using Biomart. As the distribution of transcript lengths is sharply peaked, ten quantiles of transcript length were used to group the genes for display of TTTP score distributions. To control for neuron specificity, the data on single cell transcriptomes from the Linnarsson/Hjerling-Leffler labs (Zeisel et al., 2015) was utilized (data available at linnarssonlab.org/cortex/). The cell-type specificity matrix was used to produce a metric for neuron vs glia enrichment. This was done by first calculating neuronal expression as the sum of expression in all cells they label as 'pyramidal' or 'interneurons', whilst glial expression was the sum of expression in 'astrocytes', 'endothelial', 'microglia' or 'oligodendrocytes'. The ratio of these values was taken, and classic markers checked to ensure they were as expected (Dlg4 = 15; Camk2a = 6; Map2 = 13; Gfap = 0.2; Mbp = 0.1; Aif1 = 0.2). The 5000 genes most enriched for neurons were then used to perform an ALiGeT analysis for the combined Schizophrenia list. To confirm that the threshold used for how neuron-specific the genes are does not influence this result, we also show below a figure with PeGeT results using different thresholds between 1000 and 8000 (Figure 6—figure supplement 1).

To perform the analyses restricted to up/down-regulating genes depicted in Figure 6c,d all probes which show a higher/lower expression level at the beginning of the spline, relative to the splines value at the TTTPs are dropped. Removal of duplicate probes targeting the same HGNC gene is performed after dropping the probes. Probes which do not have TTTPs are also dropped from from both analyses.

## Functional analysis of genes with high PeGeT scores

For the comparisons of mouse and human genes with large PeGeT scores, orthologs were determined using the Biomart package, through querying which HGNC Genes are orthologs for a given MGI ID. To find GO terms enriched in this set of genes, the MGI symbols were analysed using DAVID (Huang et al., 2009) against a background of mouse genes, and the GO term for 'synaptic transmission' was found to be the most enriched (the Benjamini-Hochberg corrected p-value is stated in the text). For the analysis of phenotypes from the Mammalian Phenotype Ontology (Smith and Eppig, 2009) (MPO), a full database of phenotypes was downloaded from ftp.informatics.jax.org. Any genes which were not detected as present by the microarrays were dropped from the MPO database, and a hypergeometric test for significance of enrichment performed on the remainder. Enrichment for 'behavioural' and 'nervous system' phenotypes was specifically tested for, and hence probabilities presented are not adjusted for multiple hypothesis testing.

## DEGET and ALIS using genome wide summary statistics

Gene association statistics were calculated from Genome Wide Association Study (GWAS) Summary Statistics using v1.05 of MAGMA (Multi-marker Analysis of GenoMic Annotation) (de Leeuw et al., 2015). MAGMA enables disease/phenotypes association scores to be calculated for each gene,

while accounting for linkage disequilibrium and the contributions from multiple SNPs. MAGMA takes GWAS summary statistics as input: these do not contain data on the individuals from the study and simply list the association p-values and z-scores/odds ratios for each SNP included in the study. The 1000 Genomes European panel was provided to MAGMA as reference data for calculating Linkage Disequilibrium. The schizophrenia summary statistics are associated with the GWAS analysis performed by the Psychiatric Genomics Consortium (PGC) which found 108 genome wide significant loci (*Schizophrenia Working Group of the Psychiatric Genomics Consortium, 2014*); the file was downloaded from the PGC website (https://www.med.unc.edu/pgc/results-and-downloads).

DEGET enrichment was calculated by first grouping the genes into deciles based on the age of the turning point. DEGET groups for the MAGMA method are different from those calculated for gene list approaches for several reasons: firstly, sorting is performed using entrez gene IDs rather than HGNC gene symbols, secondly, all genes within the extended MHC region are removed from these analysis (as their MAGMA gene associations cannot be properly assigned due to Linkage Disequilibrium in this region). DEGET scores were assigned to genes as they were done for the gene set based analysis. To determine whether a given GWAS trait is enriched at a particular age, the z-score calculated by MAGMA for each gene was multiplied by the DEGET score. Z-scores were then shuffled 20,000 times, multiplying at each iteration with the DEGET scores, and compared to the unshuffled value. The p-value is based on the frequency with which the sum of unshuffled value is greater than the shuffled values. The same approach (multiplying ALIS scores with MAGMA z-scores, following by perturbations of z-scores) is used to calculate ALIS enrichment probabilities.

## Age prediction

All of the human (including fetal) and all mouse samples were included in the age prediction analysis. Age predictions were performed with radial basis function Support Vector Machines through the e1071 package that provides an interface to libsvm in R (*Chang and Lin, 2011*). Two rounds of 10-fold partitioning were used to form training, validation and test sets. An initial round of random partitions separated test data from training/validation data. The combined set of training/validation data was then passed to the *tune.svm()* function from the e1071 package which then uses 10-fold cross-validation to perform a parameter search. A first shallow grid search was performed for $\gamma \in -15, -13, \ldots, 1\ 3)$ and $cost \in -5, -3, \ldots, 13, 15)$, and the optimal pair of values selected as $(\gamma_1, cost_1)$. A second finer grid search was then performed over $\gamma \in \gamma_1 - 1.5, \gamma_1 - 1.375, \ldots, \gamma_1 + 1.375, \gamma_1 + 1.5)$ and $cost \in cost_1 - 1.5, \ cost_1 - 1.375, \ldots, cost_1 + 1.375, cost_1 + 1.5)$.

Probes were included in the model by first determining which probes are associated with age, as determined using a linear model. The linear modeling of age-associated probes was repeated for each of the partitioned sets of training/validation data; as such, data in the test set did not influence the selection of probes through the linear model. The probes were then ranked based on the level of association, and the top $N_{cutoff}$ probes were used for training and testing. The reported accuracies were obtained using values of $N_{cutoff}$ 40 for humans and 100 for mice. To determine the appropriate level of $N_{cutoff}$, a range of values were tested between 10 and 400, and the optimal number manually selected on the basis of having optimal SSE without using an unnecessarily large number of probes.

## PSD preparations for mass spectrometry

Crude PSD preparations were made from dissected mouse forebrain tissue from C57BL6/5J mice ages 1 month to 5 months. In brief, each forebrain was homogenized by performing 12 strokes with a Dounce homogenizer containing 5 mL of ice cold homogenization buffer (320 mM sucrose, 1 mM HEPES, pH = 7.4) containing 1X Complete EDTA-free protease inhibitor (Roche) and 1X Phosphatase inhibitor cocktail set II (Calbiochem). Insoluble material was pelleted by centrifugation at 1000 x g for 10 min at 4°C. The supernatant (S1) was removed and the pellet was re-suspended in 2 mL of homogenization buffer and an additional six strokes were performed. Following a second centrifugation at 1000 x g for 10 min at 4°C, the supernatant (S2) was removed and pooled with S1. The combined supernatants were then centrifuged at 18, 500 x g for 15 min at 4°C. The pellet was re-suspended in 5 mL of extraction buffer (50 mM NaCl, 1% DOC, 25 mM Tris-HCl, pH 8.0) containing 1X Complete EDTA-free protease inhibitor (Roche) and 1X Phosphatase inhibitor cocktail set II

(Calbiochem) and incubated on ice for 1 hr. The resulting crude PSD extracts were centrifuged at 10,000 x g for 20 min at 4 ℃ and the resulting supernatant filtered through a 0.2 µm syringe filter (Millipore).

## Preparation of samples for LC-MS/MS

Protein concentration of PSD preparations was determined using 1X Quickstart Bradford assay (Bio-Rad). Thirty micrograms of PSD protein was prepared to contain 1X Novex NuPAGE LDS sample loading buffer (Invitrogen) with 100 mM DTT (BioRad), boiled for 10 min at 100℃ and loaded into 1 well of a 10-well Novex NuPAGE 3–12% Bis-Tris gradient gel (Invitrogen). Electrophoresis was performed under reducing conditions using the Novex NuPAGE SDS-PAGE system (Invitrogen) for 5 min. Gels were then stained with SimplyBlue SafeStain (Invitrogen) following the manufacturer's instructions. Gel bands were excised and subjected to tryptic digestion using standard methods.

## Mass spectrometry and data analysis

Five microgram of tryptic digest was analysed by LC-MS/MS using a UPLC Dionex QExactive (Thermo-Fisher, Waltham, Massachussets, USA). Protein identification was performed with MASCOT (Matrix Sciences) using Uniprot Mouse database (downloaded on 2014 March 24th). Label-free quantitation analysis was then performed on all timepoints examined in the study using Progenesis (Nonlinear Dynamics). The normalized Mass Spectroscopy dataset was read into R. Swissprot/Trembl ID's for each detected protein was matched to a gene symbol. Orthologs were determined using the Biomart package, through querying which HGNC Genes are orthologs for a given MGI ID. The data was $\log_2$ transformed. A linear model was fitted to estimate protein expression based on age and sex using the bioconductor 'l*umi*' package. Significance of differential expression was corrected using the Benjamini-Hochberg method.

Build #85 of the Human Phenotype Ontology was downloaded from http://compbio.charite.de/hudson/job/hpo.annotations.monthly/lastStableBuild. All diseases associated with the following HPO terms were extracted from the ontology using Phenexplorer (compbio.charite.de/phenexplorer): 'Neoplasm', 'Abnormality of the Nervous System', 'Abnormal peripheral nervous system morphology' and 'Abnormality of the autonomic nervous system'. Using these lists all diseases associated with neoplasms, and peripheral/autonomic disorders, and which are not associated with nervous system disorders, were dropped from the HPO phenotype database. From the remaining dataset, congenital onset disorders were taken as those annotated with the following terms: 'Congenital onset', 'Infantile onset' or 'Neonatal onset'. Adult onset disorders were annotated with either 'Adult onset', 'Young adult onset', 'Schizophrenia' or 'Bipolar affective disorder'. Schizophrenia enrichment was tested using the combined list described earlier. Enrichment analyses were performed using hypergeometric tests. Functional analysis was done by manually assigning a single functional category to each of the proteins detected in the dataset. Functional annotations were based on those from a previous paper on the synaptic proteome (*Emes et al., 2008*) and expanded based on Panther protein class.

## Code availability

R code was used for all TTTP and age prediction analyses described herein (see Bioconductor, 'TurningPoints').

## Acknowledgements

Support from the Medical Research Council, Wellcome Trust, European Union Seventh Framework Programme (FP7 grant agreement no. 242167). T Le Bihan and L Imrie at SynthSys, University of Edinburgh for mass spectrometry sample analysis. The LC-MS QExactive equipment was purchased by a Wellcome Trust Institutional Strategic Support Fund and a strategic award from the Wellcome Trust for the Centre for Immunity, Infection and Evolution (095831/Z/11/Z). D Maizels for artwork.

## Additional information

### Funding

| Funder | Grant reference number | Author |
|---|---|---|
| Seventh Framework Programme | HEALTH-F2-2009-241995 | Seth GN Grant |
| Seventh Framework Programme | HEALTH-F2-2009-242167 | Seth GN Grant |
| Wellcome Trust | Graduate Student Fellowship | Nathan G Skene |
| Wellcome Trust Sanger Institute | Core funding | Seth GN Grant |

The funders had no role in study design, data collection and interpretation, or the decision to submit the work for publication.

### Author contributions
Nathan G Skene, Data curation, Software, Formal analysis, Investigation, Methodology, Writing—original draft, Writing—review and editing; Marcia Roy, Data curation, Formal analysis; Seth GN Grant, Conceptualization, Supervision, Funding acquisition, Writing—original draft, Project administration, Writing—review and editing

### Author ORCIDs
Nathan G Skene, http://orcid.org/0000-0002-6807-3180
Marcia Roy, http://orcid.org/0000-0002-4088-3958
Seth GN Grant, http://orcid.org/0000-0001-8732-8735

### Ethics
Animal experimentation: All animal experiments conformed to the British Home Office Regulations (Animal Scientific Procedures Act 1986; Project License PPL80/2,337 to Prof Seth Grant), local ethical approval, and NIH guidelines.

### Decision letter and Author response
Decision letter https://doi.org/10.7554/eLife.17915.036
Author response https://doi.org/10.7554/eLife.17915.037

## Additional files

### Supplementary files
• Supplementary file 1. (a) Human PEGET scores. (b) Mouse PEGET scores. (c) Set of genes that were found to have the top PEGET scores in both human and mouse. (d) Post-synaptic proteome gene sets. (e) Annotated table of proteins differentially expressed with age as well as the assigned functional classes used to determine enrichments. (f) Disease gene sets
DOI: https://doi.org/10.7554/eLife.17915.026
• Transparent reporting form
DOI: https://doi.org/10.7554/eLife.17915.027

### Major datasets
The following dataset was generated:

| Author(s) | Year | Dataset title | Dataset URL | Database, license, and accessibility information |
|---|---|---|---|---|
| Nathan G Skene, Marcia Roy, Seth GN Grant | 2015 | Transcription profiling by array of wild type mouse hippocampi from 186 animals of both genders and two background strains (C57Bl/6 and 129s5) between the ages of 58 to 600 days | https://www.ebi.ac.uk/ar-rayexpress/experiments/E-MTAB-3256/ | Publicly Available at EBI ArrayExpress (accession no: E-MTAB-3256) |

The following previously published datasets were used:

| Author(s) | Year | Dataset title | Dataset URL | Database, license, and accessibility information |
|---|---|---|---|---|
| Colantuoni C, Lipska BK, Ye T, Hyde TM, Tao R, Leek JT, Colantuoni EA, Elkahloun AG, Herman MM, Weinberger DR, Jaffe A, Kleinman JE | 2011 | Temporal Dynamics and Genetic Control of Transcription in the Human Prefrontal Cortex | http://www.ncbi.nlm.nih.gov/geo/query/acc.cgi?acc=GSE30272 | Publicly available at the NCBI Gene Expression Omnibus (accession no: GSE30272) |
| Kang HJ, Kawasawa YI, Cheng F, Zhu Y, Xu X, Li M, et al | 2011 | Brainspan: Atlas of the developing brain | http://www.brainspan.org/static/download.html | NA |
| Somel M, Franz H, Giger T, Nickel B, Bahn S, Webster MJ, Weickert CS, Lachmann M, Pääbo S, Khaitovich P | 2009 | Gene expression changes during primate postnatal brain development | http://www.ncbi.nlm.nih.gov/geo/query/acc.cgi?acc=GSE11512 | Publicly available at the NCBI Gene Expression Omnibus (accession no: GSE11512) |

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
