## [Decision Letter]

Thank you for submitting your article "A genomic ageing program that reorganises the young adult brain is targeted in schizophrenia and anxiety disorders" for consideration by *eLife*. Your article has been reviewed by two peer reviewers, and the evaluation has been overseen by a Reviewing Editor and a Senior Editor. The reviewers have opted to remain anonymous.

The reviewers have discussed the reviews with one another and the Reviewing Editor has drafted this decision to help you prepare a revised submission.

Summary:

The authors create a new method to quantify transcriptome trajectory turning points (TTTP) and apply this to human BrainCloud dataset and a new mouse brain transcriptome dataset across ages 58-600 days. The authors identify a peak age of TTTP around 25-30 years old, which they link to psychiatric disorders (schizophrenia and autism) and synaptic development. Overall the reviewers felt that this was an ambitious analysis covering a novel and interesting topic (complex brain gene expression trajectories).

Essential revisions:

1) The text needs attention. The Introduction is long and meandering and would be more accessible if shortened significantly. The authors need to explain what they have done with greater clarity. For example the model the authors propose (subsection “Transcriptome trajectories and schizophrenia” paragraph three) is so vague as to be nearly meaningless.

2) The authors need to explain better which models they used and why. There are several algorithms presented in this work (TTTP, peget, DeGET, AliGET), of which only TTTP is straightforward. It is hard to understand why the authors choose to use one over the other. Some of the models (such as Aliget) include parameters that do not have a clear biological justification and seem somewhat arbitrary. Comparison of units across species is also unclear – is a human "year" equivalent to a mouse "day"?

3) Correction for multiple testing needs to be made explicit. For the cell type analysis a Bonferoni correction is used, but it is not clear that appropriate corrections were made for other analyses (the multiple different enrichment tests).

4) A list of >600 de novo mutation-containing genes is used. However, this is not a list of genes with any level of reasonable statistical support – what is the FDR? Are these all mutations in genes never observed in controls? The anxiety gene list is not well statistically supported and has no place in such an analysis.

5) Some of the models include a δ-E term, which "negates the contribution of genes that merely fluctuate away from their mean." The δ-E term also likely biases analysis toward genes with certain levels of baseline expression, as well as the dynamic range of microarrays. In addition, the gene fluctuation issue should be adequately accounted for by using spline regressions and biological replicates.

6) With regard to the age classifier, it is unclear if this is developed using a test set and validation set, which is necessary to truly evaluate accuracy that would be generalizable and meaningful. If it is not, they must find an independent validation set to make this of any meaning. In this regard, the cell type analysis does use a validation set for example.

7) All models are based on the TTTP point, which is defined as the earliest age in which the first derivative of expression changes signs. It is not clear why the authors only use the earliest of such points when genes can clearly have multiple inflections across the lifetime (see Figure 2—figure supplement 1). By taking the earliest inflection point, this makes the analysis somewhat contingent on youngest sample in a dataset (which is not balanced between mouse and human experiments), as well as the resolution of samples at the younger timepoints, which is sparse for BrainCloud (see below). Finally, all models are highly contingent on (18) the quality and resolution of the input datasets and (22) the accuracy of the spline regression model. As shown in the supplement, choice of regression parameters can have dramatic effects on downstream analyses. There is no justification for the cubic spline model used or any assessment of the accuracy with which it fits the underlying data. The choice of parameters seems arbitrary.

8) Gender differences – Using the Braincloud dataset, the peak age of TTTP is reported to be delayed in females compared to males and this is used as an explanation for why psychiatric diseases occur later in F. However, the Braincloud data that this result is based on has already had sex effects regressed out prior to analysis. It is inappropriate to make any comment on sex effect without using the non-regressed dataset.

9) Braincloud is used to define the human TTTP trajectory and includes human prenatal and postnatal samples. The regressed and normalized data is downloaded from GEO. However, it is unclear whether measures of RNA quality and other sources of technical variability are taken into account in any of the analyses. There will likely be an interaction between age and RNA quality and therefore potentially more variability in gene expression signal seen during certain timepoints. Replication in the Kang et al. (GSE25219) dataset will be important as this is similar to BrainCloud in scope.

10) The authors used the cubic splines with 3 DOF to depict the gene expression trajectories, and the TTTPs were detected by searching slope changing points along the smooth curves. How well does this actually fit the data? Why were 3 DOF chosen? There should be some rigorous assessment of accuracy of regression model (compared with Loess, etc)

11) The degree of smooth curve and the number of binned intervals could affect the number of identified TTTPs. But in subsection “Detecting turning points (TTTPs” the author said "If multiple turning points were found in a single spline, then only the first was used". This strategy may result in the imprecisely predicting TTTPs, e.g., the differences shown in Figure 4—figure supplement 2 when using different parameters and different methods.

12) Human-Mouse comparisons: the authors find a peak of TTTP around age 160 days in mice which they imply is equivalent to 26-30 years in human. However, mice were not assessed before 58 days postnatally whereas human time-points include prenatal brain samples. Since TTTP is calculated as the earliest transition point (if multiple are detected in the sample gene) it is likely sensitive to the age windows assessed.

13) Age differences – the peak age of TTTP is reported to be in the range of 25-30 years in human. However, there are a relatively few number of samples in the 2-20 age groups compared with age 0 and 20+. The ability to identity changes in transcriptomic trajectory during this crucial period of brain development is strongly unpowered and could confound the age trajectory of TTTP.

14) Psychiatric enrichments – subsection “Psychiatric susceptibility genes in young adults”, paragraph two: "TTTP.…accurately predicted the age windows for the onset of schizophrenia and anxiety disorders." As shown in Figure 5—figure supplement 3, this is highly contingent on parameter choice of the model used to fit transcriptome data. Loess regression or cubic spline with 4 dof predicts much earlier onset of schizophrenia (~16 year old). The choice of parameters does not seem to be justified either biologically or in terms of fit to the dataset. As such, this conclusion cannot be considered robust.

15) Cell-type enrichment analyses are likely on very small numbers of genes, which is susceptible to bias. The Zeisel dataset was used to define single cell transcriptomes. However, this dataset is based on single-cell RNAseq with <3000 genes expressed in each cell. Genes were dropped from analysis that were not expressed in Zeisel and therefore the input set of genes is likely very small.

16) The author claimed the accuracies of age prediction are 6.5 years in human and 28 days in mice. If this prediction included fetal samples, the prediction is imprecise. One similar age prediction from DNA methylation reported the predicting difference is close to zero for embryonic samples (Genome Biology 2013, 14:R115). If this prediction did not include fetal samples, the authors should not state that, "Remarkably, they showed accurate age predictions across the entire range of ages in both species," as Braincloud includes 38 fetal samples. Similarly, the authors should make clear that they excluded these samples, and, in light of the dramatically higher temporal dynamics observed in fetal and infant periods in these samples (Nature 478, 519-523, Figure 2), why they excluded these samples.

[Editors' note: further revisions were requested prior to acceptance, as described below.]

Thank you for resubmitting your work entitled "A genomic ageing program that reorganises the young adult brain is targeted in schizophrenia and anxiety disorders" for further consideration at *eLife*. Your revised article has been favorably evaluated by a Senior editor, a Reviewing editor, and one reviewer.

The manuscript has been improved but there are some remaining issues that need to be addressed before acceptance, as outlined below:

*Reviewer #1:*

In my original review, I was very enthusiastic about the innovative approach and ideas contained in this paper, but was concerned about Methods and potential confounders, which were hard to understand and evaluate in the original paper. The other reviewer had very similar concerns. The authors now provide a detailed response to the critiques and a revised manuscript, which is much improved. They have rewritten, shortened many sections and improved clarity. They have done a good job of clarifying parameter choice and for some of algorithms, and shown the robustness of the methods to these choices.

I should emphasize that I still have my original enthusiasm about this innovative study. But although some of my concerns are well addressed, others regarding the robustness of the method and interpretation of the data are not and still need additional work. Because this study could illuminate some areas of disease susceptibility that have been mysterious, it could have high impact, but it is critical that it be methodologically solid in every respect. I think it is now believable that young adult is a time of interesting changes in gene expression trajectories, although as the authors acknowledge choice of spline parameters effects the actual age considerably. This is not a minor concern that needs acknowledgement, but a key point of their analysis. Better if the authors presented an interval of TTTP ranges based on parameter choices just to show what we can be certain about from these data. The major question that still remains is how is this related to disease specific factors.

Other concerns are as follows:

1) The main focus on the paper is defining transcripome trajectory turning points, which they then relate to disease. They have clarified the use of algorithms and Methods. Now that some of these issues are clarified, one major question still remains. If the TTTP indeed does define SZ onset at late adolescence with a peak at 25 years (although with some range depending on parameters chosen as mentioned above), it should be specific to schizophrenia and not to other disorders. The choice of gene sets to be compared is critical here, and in no way are the genetic studies of the brain disorders that are compared comparable. Some identify mostly common variants of small effect size, and others, such as ID and autism, mostly rare variants with large effect size. These mutations are under considerably different selection pressures as their effects sizes differ.

So, a major problem that still remains is that the choice of the 600 SZ de novo genes is not well supported by statistical genetic analysis. The authors’ response to this in the revision is not satisfactory. There is strong support for de novo mutations as a class in SZ and other neurodevelopmental disorders, and some pathways, more equivocally. But, the individual genes in this list are mostly not supported by evidence yet. It is not a question of finding mutations that occur only in SZ patients and not in controls; it is standard population genetics. de novo mutations occur frequently and many of these genes are almost certainly not risk genes. The authors should absorb some of the analyses in the following papers and should filter the list by metrics that are now well accepted in human genetics (PMID:27899611; PMID:25086666; PMID: 27535533; PMID:27533299; PMID:26439716; PMID:25684150).

Also in this regard, genome-wide significant genes for anxiety disorders have not been identified. Again, the use of a hodge podge of gene lists, without strong statistical support or rationale plagues this major aspect of the paper. The choice of gene lists is so important that it should be presented up-front and clarified. And when comparisons are made with these lists, they should be of equal power, or they don't support disease specificity. In other words, the gene lists should be the same size approximately, or better yet, account for a similar proportion of the population attributable genetic liability to account for the different effect sizes of mutations and common variants. Further, anxiety should certainly not be used at all, unless the authors can provide genome-wide statistical support for the genes identified.

If the authors want to use this SZ list of 600 genes that lacks statistical support or OR for most of the genes, they should definitely compare to a similarly filtered of genes that cause autism and intellectual disability, or list of brain enriched genes (2/3 fold?) to see if the trajectory that they see in SZ is meaningful, as mentioned above. In the gene lists there are 2- 3x more SZ genes than ASD genes (one list has 170, the other about 380), and there are only about 80 ID genes listed, when there are >400 known. At least the sensitivity of the TTTP analyses to the size of the gene list should be provided. Another ground truth would be to show that a similar sized list of PSD genes, or highly synaptically enriched genes not chosen for association with SZ did not show the same pattern. It simply may be that certain classes of synaptic genes are enriched for this pattern of trajectory switching and this creates a vulnerable period, but it is not due to an enrichment of SZ risk genes themselves.

3) The human-mouse comparison is really a tough area as the authors recognize, since it has been demonstrated that there is not a linear, constant scaling of stage comparability between these species across development when specific developmental processes are compared. The authors should at least cite some of the work that demonstrates this and discuss it more thoroughly as a caveat. Currently it is somewhat glib. They have done a very good job of dealing with this issue analytically, but the readers should know that it is a biological conundrum that is tough to address with any method mapping human development to mouse, and what the limitations are.

---

## [Author Response]

*Essential revisions:*
*1) The text needs attention. The Introduction is long and meandering and would be more accessible if shortened significantly. The authors need to explain what they have done with greater clarity. For example the model the authors propose (subsection “Transcriptome trajectories and schizophrenia” paragraph three) is so vague as to be nearly meaningless.*

We acknowledge that the text would benefit from improvement in clarity and brevity. We have edited the Abstract, and significantly shortened the Introduction and Discussion. We have also rewritten the model and introduced the phrase “genetic lifespan calendar” to help convey that the gene regulatory events occur at particular ages.

*2) The authors need to explain better which models they used and why. There are several algorithms presented in this work (TTTP, peget, DeGET, AliGET), of which only TTTP is straightforward. It is hard to understand why the authors choose to use one over the other.*

We have improved the clarity of the explanation of these approaches and how they complement one another in the first paragraph of the Results. Where we can do so without overwhelming the reader we have provided ALiGeT, DeGeT and PeGeT all together.

*Some of the models (such as Aliget) include parameters that do not have a clear biological justification and seem somewhat arbitrary.*

There is one parameter used in the human ALiGeT calculation: the base of the exponential is set as 1.5 (the one other parameter used for the mouse dataset is explained below in response 2.iii). This parameter sets the distance over which genes can have a strong influence on the ALiGeT score for a given year. It has a fairly direct biological interpretation: larger values imply that functionally related sets of genes will ‘turn’ in increasingly small windows of time and that the dataset is able to accurately detect the turning points. This parameter is explained in detail in subsection “Scoring TTTPs”. We show in Figure 5—figure supplement 5 that the choice of parameter does not have a particularly significant effect on the results. We now reference this figure supplement in the Materials and methods section.

*Comparison of units across species is also unclear – is a human "year" equivalent to a mouse "day"?*

We inadvertently had left these details out of the methods and thank the reviewer for bringing it to our attention. We have amended the Materials and Methods section to make this clear.

For the ALiGeT analysis 8.33 mouse days is equivalent to one human year. This was calculated on the basis that the species can be considered in old age at 650 days (for mice) and 78 years (for human). Then 650/78=8.33.The term in the ALiGeT function where length of time enters the equation is in the exponent. We thus scale the term by dividing by 8.33. The width of the ALiGeT decay function is then approximately equal (in proportion of lifespan) between the two species.

*3) Correction for multiple testing needs to be made explicit. For the cell type analysis a Bonferoni correction is used, but it is not clear that appropriate corrections were made for other analyses (the multiple different enrichment tests).*

We have made the use of multiple testing correction more explicit in the paper. Where a p-value is Bonferroni corrected we now denote it as. Where a p-value is Benjamini-Hochberg corrected with FDR=0.5 we now denote it as. The Materials and methods section now makes clear for every analysis performed whether and how multiple testing correction was performed.

*4) A list of >600* de novo *mutation-containing genes is used. However, this is not a list of genes with any level of reasonable statistical support – what is the FDR? Are these all mutations in genes never observed in controls?*

To the best of our knowledge there is not a single mutation associated with schizophrenia—whether a genome wide significant SNP or de novo deletion—which is never found in controls. The schizophrenia associated mutations with highest penetrance are a set of copy number variations which generally occur as a result of de novo mutations. There is strong statistical support for this (see for instance PMID: 21855053). de novo mutations have been found to occur at a higher rate in schizophrenics relative to controls (PMID: 23911319). Those de novo mutations which do occur are found at very significantly enriched rates in particular groups of genes, such as those involved in NMDA receptor function (PMID: 24463507). The functional associations found for de novo mutations are the same as for those found through GWAS studies. We conclude that there is strong statistical support for de novo mutations playing a role in Schizophrenia, and that the set of genes found to bear mutations acts in schizophrenia related functional pathways.

We further note that we do already perform separate analyses for genes found through GWAS and de novo studies and find the same results (see Figure 5). A result that was important for us is that we used two separate sets of de novo genes (treating those from Gulsuner et al. as separate) and again find the same result for both (Figure 5—figure supplement 6).

*The anxiety gene list is not well statistically supported and has no place in such an analysis.*

We agree that the anxiety associated genes are not ideal, however we believe that it is reasonable to use them with the appropriate caveats. Preferably we would have used separate genome wide significant genes lists for each anxiety disorder subtype, but it is likely to be many years before this data becomes available. The gene lists that we used combines knowledge from all available sources that could be brought to bear on the issue, including human and mouse genetics. We have made it clear that the human genetic basis for the anxiety gene set is weaker than that of others disorders. See subsection “Psychiatric susceptibility genes in young adults; “It should be noted that the human genetic evidence for the anxiety gene set is not as substantial as that for schizophrenia and included mouse genetic data (see Materials and methods). We expect these results to be further evaluated with larger datasets including those for specific anxiety disorders in the future.” In a discussion paragraph where we discuss the relationship between schizophrenia and anxiety, we again remind the reader of this point: “The genetic basis of anxiety disorders is not as well characterised as schizophrenia and we expect that future large-scale datasets for specific anxiety disorders with adolescent onset will enhance our understanding of the role of the genetic calendar.” We believe this to be reasonable and provides the readers with the appropriate information for them to judge.

5) Some of the models include a δ-E term, which "negates the contribution of genes that merely fluctuate away from their mean." The δ-E term also likely biases analysis toward genes with certain levels of baseline expression, as well as the dynamic range of microarrays.

We acknowledge that the phrasing of that sentence made it sound as though the primary purpose of the δ-E term was “to negates the contribution from genes that merely fluctuate away from their mean”. We have now rewritten that section (“Quantifying expression trajectoris”) to explain that the δ-E term is fundamental to this analysis as it focuses the turning point analysis on those genes with the largest role in brain development.

As the core results have been replicated using the BrainSpan RNA-Seq dataset (Figure 4—figure supplement 1 and Figure 5—figure supplement 2) it is clear that the dynamic range of microarrays does not influence our findings.

*In addition, the gene fluctuation issue should be adequately accounted for by using spline regressions and biological replicates.*

It is unclear how “spline regressions and biological replicates” could ever stop TTTPs with very small ΔE terms from occurring: if a gene has stable expression it will show (very) small changes in direction at some point during the lifespan due to random fluctuations. Only genes which consistently change in one direction will avoid this. This can be proved by taking random samples from a normal distribution and fitting splines to the sampled values—at some point the spline will change direction due to random effects. Thus small values of ΔE indicate real biological information: that the gene’s expression is stable.

*6) With regard to the age classifier, it is unclear if this is developed using a test set and validation set, which is necessary to truly evaluate accuracy that would be generalizable and meaningful. If it is not, they must find an independent validation set to make this of any meaning. In this regard, the cell type analysis does use a validation set for example.*

We can confirm that we partitioned the data into training, validation and test data sets. This was done through two separate rounds of 10-fold cross-validation. The grid search was performed using the tune.svm() function from the e1071 package—this performs 10-fold crossvalidation and was thus passed both the training and validation datasets (separate from the test data). The relevant part of the Materials and Methods section has been improved to make this clearer. We have made it more explicit in the Results section that the data is partitioned into separate datasets sets.

*7) All models are based on the TTTP point, which is defined as the earliest age in which the first derivative of expression changes signs. It is not clear why the authors only use the earliest of such points when genes can clearly have multiple inflections across the lifetime (see Figure 2—figure supplement 1).*

In Figure 2 we show that the majority of genes plateau after the first turning point, indicating that later turning points are not as pervasive and important as the first ones.

*By taking the earliest inflection point, this makes the analysis somewhat contingent on youngest sample in a dataset (which is not balanced between mouse and human experiments), as well as the resolution of samples at the younger timepoints, which is sparse for BrainCloud (see below).*

Acknowledging this point, we have now introduced a new figure (Figure 5—figure supplement 6) which shows that our core results obtained with the Braincloud dataset are stable to removal of all fetal samples. This shows that the analysis is not overly contingent on the age of the youngest samples. We do acknowledge that the difference of starting point between human and mice may introduce differences between the two analyses: this is noted in the paper.

*Finally, all models are highly contingent on (1) the quality and resolution of the input datasets and (2) the accuracy of the spline regression model. As shown in the supplement, choice of regression parameters can have dramatic effects on downstream analyses. There is no justification for the cubic spline model used or any assessment of the accuracy with which it fits the underlying data. The choice of parameters seems arbitrary.*

We have shown in Figure 5—figure supplement 3 that the core results remain stable when we vary the spline regression parameters.

The cubic spline with 3 degrees of freedom was selected as it balanced the competing priorities of: (a) reproducing the trajectories and (b) not overfitting to potentially outlying data points around the turning points. While using four degrees of freedom or loess regression necessarily result in better fits (in terms of regression residuals) this was at the cost of being more susceptible to extreme data points around the period at which the turn occurs.

If you consider the example trajectories shown for turning points at different ages shown in Figure 2—figure supplement 1 then we think you will agree that the cubic splines capture the timing of the turning points accurately.

*8) Gender differences – Using the Braincloud dataset, the peak age of TTTP is reported to be delayed in females compared to males and this is used as an explanation for why psychiatric diseases occur later in F. However, the Braincloud data that this result is based on has already had sex effects regressed out prior to analysis. It is inappropriate to make any comment on sex effect without using the non-regressed dataset.*

Gender was regressed out as a categorical factor in a linear model. As such, within the set of female samples, it is equivalent to adding or subtracting a value to all samples. It would have no influence on the shape of the trajectory. Hence our approach was perfectly valid.

*9) Braincloud is used to define the human TTTP trajectory and includes human prenatal and postnatal samples. The regressed and normalized data is downloaded from GEO. However, it is unclear whether measures of RNA quality and other sources of technical variability are taken into account in any of the analyses. There will likely be an interaction between age and RNA quality and therefore potentially more variability in gene expression signal seen during certain timepoints. Replication in the Kang et al. (GSE25219) dataset will be important as this is similar to BrainCloud in scope.*

We are happy to say that we already do perform replication with that dataset. We refer to this as the Brainspan dataset. You will note that we also use a third dataset, the ‘Somel’ dataset, for validation. See Figure 4—figure supplement 1 and Figure 5—figure supplement 2.

*10) The authors used the cubic splines with 3 DOF to depict the gene expression trajectories, and the TTTPs were detected by searching slope changing points along the smooth curves. How well does this actually fit the data? Why were 3 DOF chosen? There should be some rigorous assessment of accuracy of regression model (compared with Loess, etc)*

We have shown in Figure 5—figure supplement 3 that the core results remain stable when we vary the spline regression parameters.

The cubic spline with 3 degrees of freedom was selected as it balanced the competing priorities of: (a) reproducing the trajectories and (b) not overfitting to potentially outlying data points around the turning points. While using four degrees of freedom or loess regression necessarily result in better fits (in terms of regression residuals) this was at the cost of being more susceptible to extreme data points around the period at which the turn occurs.

If you consider the example trajectories shown for turning points at different ages shown in Figure 2—figure supplement 1 then we think you will agree that the cubic splines capture the timing of the turning points accurately.

*11) The degree of smooth curve and the number of binned intervals could affect the number of identified TTTPs. But in subsection “Detecting turning points (TTTPs” the author said "If multiple turning points were found in a single spline, then only the first was used". This strategy may result in the imprecisely predicting TTTPs, e.g., the differences shown in Figure 4—figure supplement 2 when using different parameters and different methods.*

As was shown in the original Braincloud paper, the majority of expression change occurs during the early period of life. The point of detecting TTTPs is essentially to determine when the ‘developmental’ trajectory for a given gene ends. TTTPs which occur later in life will necessarily not be relevant to this (as the developmental trajectory for that gene will have already ended). Furthermore, in Figure 2 we show that the majority of genes plateau after the first turning point, indicating that later turning points are not as pervasive and important as the first ones. As such we do not consider it necessary/valuable to consider them within this paper.

*12) Human-Mouse comparisons: the authors find a peak of TTTP around age 160 days in mice which they imply is equivalent to 26-30 years in human. However, mice were not assessed before 58 days postnatally whereas human time-points include prenatal brain samples. Since TTTP is calculated as the earliest transition point (if multiple are detected in the sample gene) it is likely sensitive to the age windows assessed.*

We have now introduced a new figure (Figure 5—figure supplement 6), which shows that our core results obtained with the Braincloud dataset are stable to removal of all fetal samples. This shows that the analysis is not overly contingent on the age of the youngest samples. We do acknowledge that the difference of starting point between human and mice may introduce differences between the two analyses: this is noted in the paper and does not diminish the results which are found.

*13) Age differences – the peak age of TTTP is reported to be in the range of 25-30 years in human. However, there are a relatively few number of samples in the 2-20 age groups compared with age 0 and 20+. The ability to identity changes in transcriptomic trajectory during this crucial period of brain development is strongly unpowered and could confound the age trajectory of TTTP.*

Unfortunately obtaining more brains of human children and adolescents is difficult and outside the scope of this study. The dataset used has far more coverage of that period than any other dataset available. We have added a cautionary note early in the Results section to ensure the readers note how this could have affected the analysis:

“We note that although the exact ages assigned to TTTPs are sensitive to both the regression method and the dataset used, and that if more data points available between childhood and early adulthood the ages assigned to particular genes would change, it is clear that the TTTP-peak reveals a major molecular reorganisation in young adults towards the end of development.”

*14) Psychiatric enrichments – subsection “Psychiatric susceptibility genes in young adults”, paragraph two: "TTTP.…accurately predicted the age windows for the onset of schizophrenia and anxiety disorders." As shown in Figure 5—figure supplement 3, this is highly contingent on parameter choice of the model used to fit transcriptome data. Loess regression or cubic spline with 4 dof predicts much earlier onset of schizophrenia (~16 year old). The choice of parameters does not seem to be justified either biologically or in terms of fit to the dataset. As such, this conclusion cannot be considered robust.*

While the precise age at which the window of enrichment occurs is dependent on the regression model used, the temporal ordering of TTTP’s is well conserved. There is a pearson’s correlation of 0.6 between TTTPs detected with cubic splines (3 df) and those detected using loess regression (when considering the 14544 probes which account for 90% of ∆E). To acknowledge your concerns, we have added a line to the Results section which brings this to the reader’s attention:

“While validating the occurrence of the disease enrichments, this analysis also revealed that the exact age at which the turning points/windows of enrichments occur depends on the regression model used (Figure 5—figure supplement 3).”

We have also noted early on in the Results:

“We note that although the exact ages assigned to TTTPs are sensitive to both the regression method and the dataset used, and that if more data points available between childhood and early adulthood the ages assigned to particular genes would change, it is clear that the TTTP-peak reveals a major molecular reorganisation in young adults towards the end of development.”

*15) Cell-type enrichment analyses are likely on very small numbers of genes, which is susceptible to bias. The Zeisel dataset was used to define single cell transcriptomes. However, this dataset is based on single-cell RNAseq with <3000 genes expressed in each cell. Genes were dropped from analysis that were not expressed in Zeisel and therefore the input set of genes is likely very small.*

The Zeisel dataset has reads from significantly more than 3000 genes per cell. The full dataset has reads from 19965 genes. A total of 16762 genes have reads assigned to cells classed as interneurons. Similarly, 15421 genes have reads assigned to cells classed as astrocytes/ependymal cells. The cell type enrichment analysis method which we used (and which we developed) was explicitly designed to utilize information from all the genes expressed in a celltype rather than just a “small number of genes”. As such, this criticism does not apply to the analysis we performed.

*16) The author claimed the accuracies of age prediction are 6.5 years in human and 28 days in mice. If this prediction included fetal samples, the prediction is imprecise. One similar age prediction from DNA methylation reported the predicting difference is close to zero for embryonic samples (Genome Biology 2013, 14:R115). If this prediction did not include fetal samples, the authors should not state that, "Remarkably, they showed accurate age predictions across the entire range of ages in both species," as Braincloud includes 38 fetal samples. Similarly, the authors should make clear that they excluded these samples, and, in light of the dramatically higher temporal dynamics observed in fetal and infant periods in these samples (Nature 478, 519-523, Figure 2), why they excluded these samples.*

We thank the reviewer for leading us to note that we had actually excluded the younger human samples. With the younger samples (including fetal) included, the prediction accuracy was improved (R^2^=0.88 and mean absolute difference between predicted and actual age of 5.48). We have changed all relevant parts of the paper to reflect this (i.e. Figure 2 and Figure 2—figure supplement 3).

We note that we do not find it surprising that it is harder to predict the age of adults than fetal samples. It has been shown that biological age predictions using DNA methylation profiles are affected by lifestyle (PMID: 25617346, PMID: 26673150) and hence we expect this reflects real biological variation.

[Editors' note: further revisions were requested prior to acceptance, as described below.]

Reviewer #1:

*[…] I should emphasize that I still have my original enthusiasm about this innovative study. But although some of my concerns are well addressed, others regarding the robustness of the method and interpretation of the data are not and still need additional work. Because this study could illuminate some areas of disease susceptibility that have been mysterious, it could have high impact, but it is critical that it be methodologically solid in every respect. I think it is now believable that young adult is a time of interesting changes in gene expression trajectories, although as the authors acknowledge choice of spline parameters effects the actual age considerably. This is not a minor concern that needs acknowledgement, but a key point of their analysis. Better if the authors presented an interval of TTTP ranges based on parameter choices just to show what we can be certain about from these data. The major question that still remains is how is this related to disease specific factors.*

We acknowledge that we did not make it clear enough that the exact age of TTTPs is dependent on choice of spline parameters. We have now amended Figure 2 such that when the distribution of TTTPs is shown, it is immediately clear in Figure 2 that the age distribution is affected by the method of spline fitting. We note that Figure 5—figure supplement 2 (which was included previously in the paper) makes it clear how the choice of spline affects individual trajectories and ALIGET enrichments, and thereby shows that the key results are robust across methods.

*Other concerns are as follows:*
*1) The main focus on the paper is defining transcripome trajectory turning points, which they then relate to disease. […] So, a major problem that still remains is that the choice of the 600 SZ* de novo *genes is not well supported by statistical genetic analysis. The authors’ response to this in the revision is not satisfactory. There is strong support for* de novo *mutations as a class in SZ and other neurodevelopmental disorders, and some pathways, more equivocally. But, the individual genes in this list are mostly not supported by evidence yet. It is not a question of finding mutations that occur only in SZ patients and not in controls; it is standard population genetics.* de novo *mutations occur frequently and many of these genes are almost certainly not risk genes. The authors should absorb some of the analyses in the following papers and should filter the list by metrics that are now well accepted in human genetics (PMID:27899611; PMID:25086666; PMID: 27535533; PMID:27533299; PMID:26439716; PMID:25684150).*

We thank the reviewer for this comment which inspired us to incorporate a new genetic analysis into the paper which directly utilizes SNP summary statistic data from the largest publically available Schizophrenia GWAS study while accounting for linkage disequilibrium and other confounds using Multi-marker Analysis of GenoMic Annotation (MAGMA).

We explicitly acknowledge the flaws with the de novo gene set when we introduce the key findings discovered using this method:

“Much of the heritability for schizophrenia is associated with SNPs that have not reached genome-wide significance with current sample sizes^52^ (and thus were not included in our analysis thus far) and the sample sizes for de novo studies are too small to determine whether any genes found are significantly associated with disease status. We therefore adapted our methods to include a greater fraction of the SNPs associated with schizophrenia heritability by using association statistics from all SNPs regardless of whether they are genome wide significant (as defined in GWAS summary statistics files) and explicitly modelling linkage disequilibrium (based on results of the 1000 genomes project), such that disease association scores can be ascertained for each gene. The ALiGeT and DEGET approaches were extended to directly utilise the gene association scores generated by MAGMA (Multi-marker Analysis of GenoMic Annotation)^53^ based on schizophrenia GWAS summary statistics. Using this approach, schizophrenia showed a DEGET enrichment at 26 years in the Braincloud dataset (=0.03135, Figure 5) and at 15—16 years in the Somel dataset (=0.0115, Figure 5—figure supplement 8) and corresponding enrichment windows were found using ALiGeT (Figure 5, Figure 5—figure supplement 8).”

The results generated using this method are shown in Figure 5, Figure 6 and Figure 5—figure supplement 8. We thereby reproduced all the key results of the paper which pertain to Schizophrenia in both in Braincloud and Somel datasets, with the exception of the ‘neuronal only’ enrichment which failed to replicate (negative result shown in Figure 6).

*Also in this regard, genome-wide significant genes for anxiety disorders have not been identified. Again, the use of a hodge podge of gene lists, without strong statistical support or rationale plagues this major aspect of the paper. The choice of gene lists is so important that it should be presented up-front and clarified. And when comparisons are made with these lists, they should be of equal power, or they don't support disease specificity. In other words, the gene lists should be the same size approximately, or better yet, account for a similar proportion of the population attributable genetic liability to account for the different effect sizes of mutations and common variants. Further, anxiety should certainly not be used at all, unless the authors can provide genome-wide statistical support for the genes identified.*

We acknowledge that the anxiety gene set did not have a sufficient level of genetic support and have dropped all results pertaining to disorder and edited the text and figures accordingly.

Please note the improvements we have made to the genetic analysis of Schizophrenia (described in response 3). For all the other disorders we maintain that this is the most strongly supported set of genes available for this set of major brain disorders.

We agree with the reviewer that for the purposes of *comparisons*, the lists should be of same size etc. However, we want to emphasise that we are not comparing these diseases – we are asking if the set of genes relevant to an individual disease are enriched in an age window.

We also don’t believe it is right to interpret our results as a comparison of the relevance of the calendar to diseases for several reasons. These diseases are known to have a different cellular basis and the gene expression data is obtained from whole tissue and not from the relevant individual cell types. Moreover, the transcriptome data is from the human prefrontal cortex, which is known to be important in schizophrenia, but is not the primary region of pathology in either multiple sclerosis or Parkinsons disease. So, we cannot exclude a role of the genetic calendar in these disorders until the relevant tissue is examined.

We believe that the paper would benefit from a clear statement about these issues at the point in the Results when the disease gene lists are first mentioned and in the Discussion.

In the Results we have added:

“Because these gene lists are not comparable (different size, population sample size, obtained using different technical approaches etc.) the relative importance of the genetic calendar to schizophrenia cannot be directly compared with these disorders (see Discussion). In addition, because the transcriptome data is from prefrontal cortex and the primary pathology of several of these diseases is in other parts of the nervous system it cannot be assumed that the transcriptome trajectories in one part of the brain are the same as in others. Hence, the lack of any detectable age window does not preclude a role for gene regulation in the onset of these diseases.”

In the Discussion we have added:

“Our studies relied on human prefrontal cortex transcriptome data, which may have limited our ability to detect a role for the genetic lifespan calendar in the age-dependent onset of those diseases that are known to have primary pathology in other brain regions (e.g. Parkinson’s disease). Given the evolutionary conservation between mouse hippocampus and human prefrontal cortex, we expect other human brain regions to show a calendar of transcriptome trajectories, but with different patterns and therefore different age windows of disease gene enrichments. Moreover, while our whole tissue transcriptome analysis appeared to be sensitive to neuronal changes, we expect that future single-cell transcriptome data will provide a more detailed insight into rarer cell types and potentially reveal mechanisms relevant to the age of onset of pathology in these cells. We do not expect that the age of onset of all brain diseases will be accounted for by the genetic calendar, as it will likely depend on the importance of cell autonomous processes and exogenous factors (e.g. inflammatory processes involving microglia). It is also likely that high penetrance severe mutations will show an earlier onset and are less likely to show a dependence on the TTTPs. Weaker alleles may manifest with undetectable or subtle phenotypes at early ages and be exposed at later ages by the changes in gene expression. Thus, TTTPs would not necessarily account for intellectual disability or autism even though some of the same genes are involved with schizophrenia.”

*If the authors want to use this SZ list of 600 genes that lacks statistical support or OR for most of the genes, they should definitely compare to a similarly filtered of genes that cause autism and intellectual disability, or list of brain enriched genes (2/3 fold?) to see if the trajectory that they see in SZ is meaningful, as mentioned above. In the gene lists there are 2- 3x more SZ genes than ASD genes (one list has 170, the other about 380), and there are only about 80 ID genes listed, when there are >400 known. At least the sensitivity of the TTTP analyses to the size of the gene list should be provided. Another ground truth would be to show that a similar sized list of PSD genes, or highly synaptically enriched genes not chosen for association with SZ did not show the same pattern. It simply may be that certain classes of synaptic genes are enriched for this pattern of trajectory switching and this creates a vulnerable period, but it is not due to an enrichment of SZ risk genes themselves.*

Firstly, we now include a sensitivity analysis to the size of both the PSD and Schizophrenia gene lists in Figure 5—figure supplement 3. This reveals that for equivalently sized gene sets, the enrichment is stronger in earlier adulthood for Schizophrenia than for the PSD. We thank the reviewer for suggesting this informative control.

Secondly, the reviewer suggests we compare to a list of brain enriched genes. We note that we already test for enrichment of pyramidal neurons, interneurons and all other major cortical cell types in Figure 3. In Figure 4 we test for enrichment of the human post-synaptic density. The different cell types and proteomes show clearly distinct lifespan trajectories.

Thirdly, in further rebuttal of the point that the Schizophrenia enrichment may simply be a result of a ‘brain expressed’ enrichment, we note that Figure 6 shows that the enrichment is present even when the analysis is limited to a subset of the 5000 most neuron specific genes. Thus, even within neuronal genes, there is a clear enrichment of schizophrenia turning points during early adulthood.

Fourth, we note again that we now include a genetic analysis of the Schizophrenia enrichment performed in accordance with what we understand to be the highest standards in the field, directly utilizing GWAS summary statistics via the MAGMA analysis package (see response 3).

*3) The human-mouse comparison is really a tough area as the authors recognize, since it has been demonstrated that there is not a linear, constant scaling of stage comparability between these species across development when specific developmental processes are compared. The authors should at least cite some of the work that demonstrates this and discuss it more thoroughly as a caveat. Currently it is somewhat glib. They have done a very good job of dealing with this issue analytically, but the readers should know that it is a biological conundrum that is tough to address with any method mapping human development to mouse, and what the limitations are.*

We appreciate the reviewer’s acknowledgement of the difficultly of handling mouse/human age comparisons. We have now included a reference to the following paper, which we believe makes the best attempt to compare brain ages between species:

Workman, A. D., Charvet, C. J., Clancy, B., Darlington, R. B. & Finlay, B. L. Modeling transformations of neurodevelopmental sequences across mammalian species. Journal of Neuroscience 33, 7368-7383 (2013).

The authors of the paper referenced above created a website with a model (http://www.translatingtime.net/translate) for translating ages between species. Using their tool we obtained the following output:

“A synapse elimination and elimination event in the cortex of the Human at PC Day 7100 translates to PC Day 156 in the Mouse.”

We have added the following sentence to the Results:

“Although there is a lack of previous research on the age equivalence of early adulthood between rodents and humans, our results are concordant with estimations made using the TranslatingTime species comparison model which suggests that p156 (5 months) is equivalent to human early adulthood based on equivalent levels of cortical synaptic maturation.”